# MV-RAG: Retrieval Augmented Multiview Diffusion

## Abstract

Text-to-3D generation approaches have advanced significantly, producing high-quality and 3D-consistent outputs. However, they often fail to produce out-of-domain (OOD) or rare concepts, yielding inconsistent or inaccurate results. To this end, we propose MV-RAG, a novel text-to-3D pipeline that first retrieves relevant 2D images from a large in-the-wild 2D database and then conditions a multiview diffusion model on these images to synthesize consistent and accurate multiview outputs. Training such a retrieval-conditioned model is achieved via a novel hybrid strategy bridging structured multiview data and diverse 2D image collections. This involves training on multiview data using augmented conditioning views that simulate retrieval variance for view-specific reconstruction, alongside training on sets of retrieved real-world 2D images using a distinctive held-out view objective: the model predicts the held-out view from other views to infer 3D consistency from 2D data. We also introduce a prior-guided fusion mechanism that dynamically balances retrieval signals with the model's prior. To facilitate OOD evaluation, we introduce a new collection of challenging OOD prompts. Experiments against state-of-the-art text-to-3D, image-to-3D, and personalization baselines show that our approach significantly improves 3D consistency, photorealism, and text adherence for OOD/rare concepts, while maintaining competitive performance on standard benchmarks.

## 1 Introduction

The automated generation of 3D content from text is important for applications such as game modeling, computer animation and virtual reality. Current approaches largely leverage pre-trained 2D text-to-image diffusion models (Song et al., 2021; Ho et al., 2020) as visual and semantic priors, either via optimization or to train generative models that produce consistent multiview images—a collection of 2D views capturing the same scene from distinct camera angles. These methods yield high-quality outputs. However, they often struggle with out-of-domain (OOD) or rare prompts, producing geometrically inconsistent results (e.g., poorly rendered unseen regions) or failing to adhere to the text, hallucinating details or replacing rare concepts with common ones.

A common text-to-3D approach uses Score Distillation Sampling (SDS) (Poole et al., 2023; Lin et al., 2023; Liang et al., 2024) to optimize a 3D representation such as NeRF (Mildenhall et al., 2020) by distilling knowledge from 2D text-to-image models. While high-fidelity, SDS-based methods often inherit the 2D prior's limitations on OOD prompts, yielding flawed 3D assets. To address this, recent work (Seo et al., 2024; Chen et al., 2024) explores retrieval augmentation, incorporating existing 3D assets as geometric priors. This improves consistency for database concepts but remains limited by scale and diversity of the 3D assets. 3D personalization techniques, e.g., DreamBooth3D (Raj et al., 2023), adapt a pretrained 2D model to a specific subject using a few (3-6) images, followed by SDS. These methods capture subject-specific details but require inference-time fine-tuning and still face geometry inconsistencies inherent to SDS. Feed-forward multiview diffusion models (Shi et al., 2023b; Liu et al., 2023a; Long et al., 2024) synthesize consistent multiview images from text or images, often fine-tuned on large 3D datasets like Objaverse (Deitke et al., 2023b). This enhances 3D awareness and improves geometric consistency over 2D-lifting approaches. Still, they struggle with OOD/rare concepts due to limited coverage in both 2D priors and 3D fine-tuning data, producing outputs with reduced photorealism, inconsistent views, or poorly inferred unobserved regions.

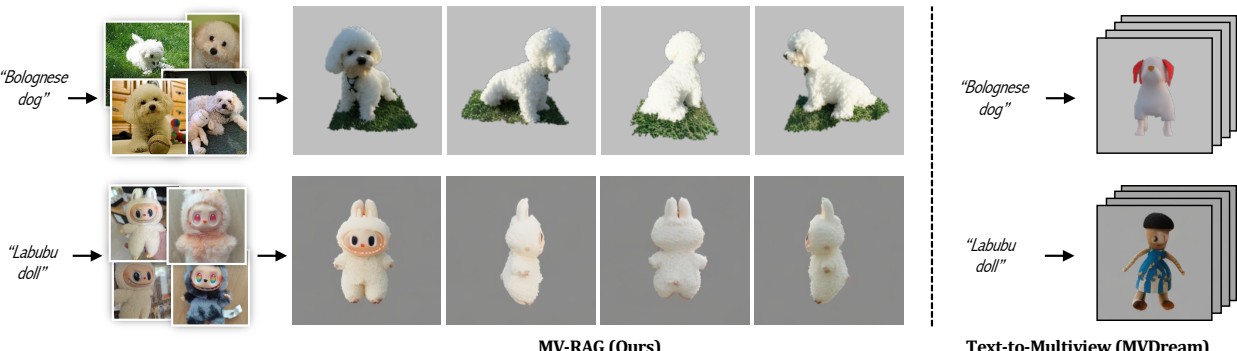

Figure 1: We introduce a retrieval-augmented diffusion framework for text-to-multiview generation. Given a text prompt, our method retrieves real-world images and adaptively leverages them together with the text, enabling faithful generation of out-of-distribution and newly emerging objects.

To this end, we propose MV-RAG, a multiview diffusion model that conditions generation on relevant in-the-wild unposed 2D images retrieved from large-scale collections. By leveraging retrieval, MV-RAG produces consistent multiview images even for rare or out-of-distribution (OOD) concepts. To enable conditioning on varying, in-the-wild views, our training combines supervision from two novel sources: (1). a *reconstruction objective* on structured multiview data, where we use augmented, "retrieval-like" conditioning views to enforce robust geometric consistency; and (2). a *hold-one-out objective* on 2D image-text data, where the model learns to infer 3D relationships by generating a held-out image from a set of $K$ related views. This hybrid scheme enables MV-RAG to learn 3D coherence while generalizing with the diverse appearance knowledge from 2D priors. To balance the influence of the base model's prior with external retrieval signals, we further introduce a fusion mechanism that dynamically adapts to the OODness of the prompt. We note that adapting RAG to the multiview setting is non-trivial, posing two key challenges requiring 3D reasoning, which are not required in the single-image RAG setting: (1) ensuring cross-view consistency when retrieved images offer only partial views, and (2) coherently composing features from different retrieved images (e.g., a car's grille from one, its wheels from another) into a single object. Crucially, applying standard injection mechanisms to unposed, in-the-wild images is fundamentally intractable without ground-truth view correspondence. Our novel hybrid training paradigm is what forces the network to infer 3D consistency from these heterogeneous inputs, unlocking the ability to use standard cross-attention adapters in a multi-view setting. Our hybrid training scheme is designed to address both (see further discussion in Appendix A.2.5).

To evaluate OOD scenarios, we curate a new benchmark of 196 challenging prompts paired with retrieved images. On this benchmark, MV-RAG significantly outperforms text-to-3D, image-to-3D, and personalization baselines in terms of 3D consistency, photorealism, and text alignment, while remaining competitive on standard in-domain benchmarks. Ablation studies further validate our design choices. An overview of our method is shown in Fig. 1.

**Contributions.** We make the following contributions: **(1)**. The first framework to successfully apply retrieval-augmented generation (RAG) to multiview 3D synthesis, achieving superior performance on out-of-distribution (OOD) concepts. **(2)**. A novel hybrid 2D-3D training scheme that bridges the gap between structured 3D data and unposed 2D image collections. **(3)**. A novel prior-guided attention mechanism that dynamically balances the model's internal prior with external retrieval signals. **(4)**. **OOD-Eval**, a new benchmark of challenging prompts to facilitate research on OOD 3D generation.

## 2 Related Work

**3D Generation Using 2D Diffusion Models** Generating 3D content by leveraging strong priors from 2D diffusion models (Ho et al., 2020) is a dominant paradigm. One major approach optimizes 3D representations, such as Neural Radiance Fields (NeRFs) (Mildenhall et al., 2020) and more recently 3D Gaussian Splatting (3DGS) (Kerbl et al., 2023), via Score Distillation Sampling (SDS) (Poole et al., 2023;

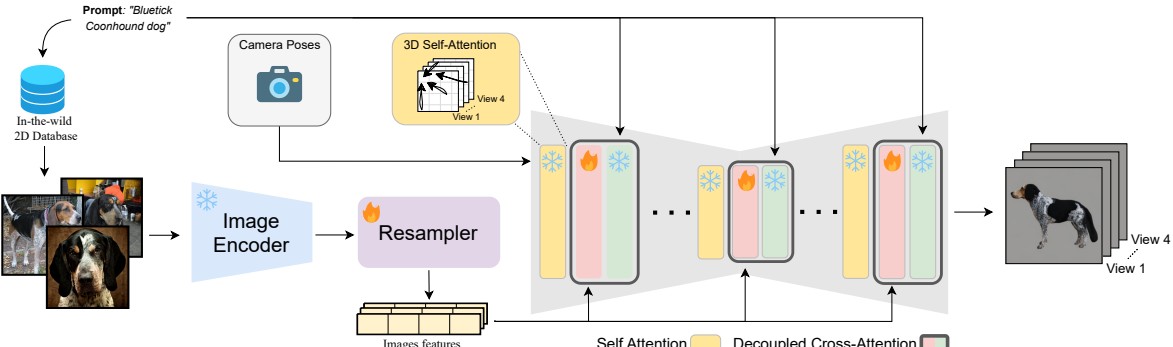

Figure 2: **Overview of our pipeline.** Given a text prompt, we retrieve $k$ relevant images from an in-the-wild 2D image corpus. Local features are extracted from each image, projected through a Resampler and integrated into retrieval-attention modules to guide the multi-view generation process.

Lin et al., 2023; Tang et al., 2024b), directly distilling knowledge from 2D priors. However, SDS struggles with geometric consistency and fidelity due to weak 3D awareness in the priors (Hong et al., 2023; Shi et al., 2023b). Feed-forward multi-view diffusion models (Shi et al., 2023b; Huang et al., 2024), often fine-tuning 2D diffusion priors with 3D dataset supervision, enhance geometric stability by directly generating multiple consistent views (Shi et al., 2023b). However, these models struggle with out-of-domain (OOD) or rare concepts due to limitations in 2D priors and insufficient 3D training data, leading to reduced photorealism or inconsistent geometry. Related image-to-multiview approaches (Liu et al., 2023a; Shi et al., 2023a; Wang & Shi, 2023; Liu et al., 2023c; Long et al., 2024), while effective with single clear inputs, are ill-suited for leveraging multiple, varied, unposed retrieved images. Our work builds on these advancements, targets OOD generation by training a multiview diffusion model to incorporate retrieved 2D images.

**Retrieval Augmented Generation (RAG)**    RAG improves generative models by incorporating external information, aiding the handling of OOD/rare entities without retraining, a successful paradigm in NLP (Lewis et al., 2020; Borgeaud et al., 2022). Notably, (Soudani et al., 2024) show that RAG is preferable to fine-tuning, especially for OOD/rare concepts. In 2D image synthesis, RAG methods similarly use retrieved images or text pairs to enhance fidelity for uncommon concepts or guide generation (Chen et al., 2022; Sheynin et al., 2022; Blattmann et al., 2022; Shalev-Arkushin et al., 2025). Recently, text-to-3D generation methods like RetDream (Seo et al., 2024) and Sculpt3D (Chen et al., 2024) retrieve existing *3D assets* for geometric priors to improve optimization consistency. However, this is limited by the scarcity and diversity of 3D databases, especially for OOD or rare concepts. Because these works leverage fundamentally different 3D infrastructures, they belong to a distinct methodological category. Inspired by findings in NLP, our work performs RAG in multiview generation via MV-RAG. By completely bypassing the 3D data bottleneck and leveraging abundant *in-the-wild 2D image datasets* to condition a multiview diffusion model, we offer a scalable way to ground OOD concept generation in real-world data.

**Personalization and 3D Congealing.**    Personalization methods adapt generative models to specific subjects using a small number of reference images, either by optimizing learned embeddings or fine-tuning model parameters (Gal et al., 2023; Ruiz et al., 2023). These ideas have been extended to 3D generation: DreamBooth3D (Raj et al., 2023) uses SDS optimization while multiview diffusion models such as MV-Dream (Shi et al., 2023b) can be fine-tuned in a DreamBooth-like manner. A key assumption in these approaches is that all input images depict the same object instance, which limits their robustness when presented with variations in structure or appearance.

Relaxing this assumption leads to a significantly harder problem. 3D Congealing (Zhang et al., 2024b) learns a shared canonical 3D representation from unposed, in-the-wild images by jointly optimizing a category-level NeRF along with per-image camera and appearance parameters. While effective, this formulation requires costly per-object optimization and often produces unrealistic textures due to input variance, highlighting the challenge of reasoning over multiple unconstrained images.

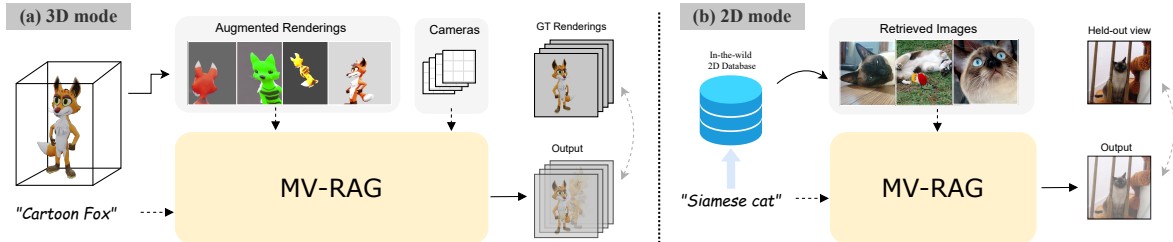

Figure 3: **Overview of our training scheme.** We adopt a hybrid training strategy that alternates between two modes. **3D mode (LHS):** A 3D object is rendered to produce ground-truth multi-view images. Additional views are generated and subjected to augmentations to serve as retrievals. These augmented views, along with the target camera parameters and associated prompt, act as input to the model. **2D mode (RHS):** We retrieve $K + 1$ images from a 2D corpus, where $K$ images are used as retrievals and one held-out image serves as the target view. In this mode, the model performs 2D self-attention rather than 3D attention, and no target camera parameters are provided.

In contrast, MV-RAG addresses this challenge through a training-time design rather than inference-time optimization. By simulating structural and appearance variance during training and integrating retrieved 2D image sets directly into a multiview diffusion model. This approach enables fast generation at inference time and leverages strong generative priors for realistic texture and structure synthesis.

## 3 Method

Given an input text prompt $p$, we first retrieve relevant 2D images corresponding to $p$ from a corpus of 2D text-image pairs. Then, we use these images, along with $p$, to guide the generation process of a multi-view diffusion model. An overview of our approach is provided in Fig. 2. We begin by describing the training process, which involves a data preprocessing stage to prepare distinct conditioning and target data for 2D or 3D training modes, followed by the training process.

### 3.1 Training Data Preprocessing

Our model training leverages geometric grounding from 3D datasets and diversity from 2D datasets. For both, we prepare 2D conditioning images related to a text $p$, simulating inference-time retrieval, alongside target supervision data. Fig. 3 illustrates our separate 2D and 3D training modes.

**2D Data Mode Supervision.** Here, we utilize a large-scale 2D in-the-wild text-image dataset (images are neither posed nor aligned). For a text prompt $p_i$, we consider $K + 1$ relevant images, from which we designate $K$ as the conditioning retrieved views $\mathcal{I}_{\text{ret}} = \{I_i\}_{i=1}^{K}$, and the remaining image, $I_{\text{target}}$, serves as the target image on which the diffusion loss is computed. This process yields training samples of the form $\mathcal{D}_{2D} = \{p, \mathcal{I}_{\text{ret}}, I_{\text{target}}\}$. No ground truth camera poses are assumed.

**3D Data Mode Supervision.** For 3D data, we assume a dataset comprising text prompts and corresponding 3D object models. For each 3D object, we render a set of $N$ ground truth target views $I_{\text{target}} = \{I_i\}_{i=1}^{N}$ at $N$ camera poses $\mathcal{C}$. We follow MVDream and use 4 orthogonal camera poses. To simulate the diverse nature of retrieved conditioning images, we render the object from $K$ additional random poses followed by a sequence of random augmentations. This yields conditioning views $\mathcal{I}_{\text{ret}} = \{I_i\}_{i=1}^{K}$. We apply a combination of geometric and semantic augmentations designed to mimic in-the-wild variability and enhance generalization. Crucially, these simulated retrievals are treated as unposed, and no camera information is provided for them during training. This yields training samples of the form $\mathcal{D}_{3D} = \{p, \mathcal{C}, \mathcal{I}_{\text{ret}}, \mathcal{I}_{\text{target}}\}$. See Appendix Sec.A.3 for additional details. We note that training by conditioning on real-world 2D retrievals was suboptimal, as the retrieved instances often differed significantly from the ground-truth 3D object, creating a conflicting training signal.

### 3.2 Retrieved and Augmented Image Encoding

A key component of our approach is encoding the $K$ conditioning images into sequences of conditioning tokens. We use a frozen CLIP ViT encoder (Radford et al., 2021) to extract patch-level features $F_i = E(I_i)$ from each image $I_i$, providing rich, spatially descriptive representations beyond a global embedding. To condense this information efficiently, we apply a learnable Resampler $\Theta_R$, inspired by the Perceiver Resampler (Jaegle et al., 2021) and IP-Adapter variants (Ye et al., 2023). $\Theta_R$ maps $F_i$ to a compact set of $N_t = 16$ tokens, $T_i = \Theta_R(F_i)$, using a small set of learnable queries attending to $F_i$. These token sequences are then used to condition the diffusion model via cross-attention, balancing expressiveness with computational efficiency.

### 3.3 Retrieval-Conditioned Multiview Diffusion

The encoded tokens are then fed into a multiview diffusion model, which extends a 2D text-to-image U-Net architecture for multiview generation. Following MVDream (Shi et al., 2023b), we incorporate camera pose embeddings for geometric guidance and modify the U-Net's self-attention layers. These layers are inflated to operate jointly over features from all generated views, forming a 3D-aware self-attention mechanism that promotes multiview consistency.

While MVDream relies solely on text-based cross-attention, we replace this mechanism with a decoupled cross-attention module that incorporates encoded tokens from both the text prompt and the retrieved images. Specifically, the tokens from the conditioning images are processed by a dedicated, trainable cross-attention branch, yielding retrieval-guided features denoted as $f_{\text{ret}}$.

For this cross-attention branch, we follow the design of IP-Adapter (Ye et al., 2023), where the U-Net query features $Q_i$, generated via a shared query projection $\theta_Q$, attend separately to keys and values from the retrieved tokens $T_i$ and the text embedding. The retrieved tokens are processed through learnable projections $\theta_{K_{\text{ret}}}$ and $\theta_{V_{\text{ret}}}$ to produce $f_{\text{ret}}$, while the text embedding is processed through frozen projections $\theta_{K_{\text{txt}}}$ and $\theta_{V_{\text{txt}}}$, inherited from a pretrained diffusion model, yielding $f_{\text{txt}}$. We note that the shared query projection $\theta_Q$ is also frozen.

This results in a decoupled cross-attention mechanism. During training, we integrate the text and retrieval features as $f = \lambda f_{\text{txt}} + f_{\text{ret}}$, where $\lambda$ is a hyperparameter. Empirically, we find that small values of $\lambda$ ease the adaptation of the newly introduced retrieval branch. These text-conditioning modules are deliberately kept frozen to preserve the base model's strong prior for in-domain concepts. The dynamic trade-off between this prior and the external retrieval signal for OOD concepts is then handled by our prior-guided attention mechanism during inference (see Sec. 3.5).

### 3.4 2D and 3D Training Modes

Our full architecture is trained jointly using the two data modes described below:

**3D Data Mode: Multiview Reconstruction.**   When training with 3D samples, the model reconstructs a set of predicted views given their camera poses $\mathcal{C}$. The U-Net's self-attention layers operate across all $N$ view latents, enforcing cross-view consistency. Each target view is conditioned on its camera pose, and the text prompt $p$ provides global guidance via its features $f_{\text{txt}}$. The visual tokens aggregated from all $K$ augmented conditioning images $\mathcal{I}_{\text{ret}}$ are used to compute the retrieval attention features $f_{\text{ret}}$, jointly guiding the generation of all $N$ target views. A multiview reconstruction loss, $\mathcal{L}_{MV}(\theta, p, \mathcal{C}, \mathcal{I}_{\text{ret}}, \mathcal{I}_{\text{pred}})$, is applied across all target views. Critically, the conditioning views $\mathcal{I}_{\text{ret}}$ simulate in-the-wild retrieval scenarios where images may share geometry but have different textures or vice versa. By reconstructing a canonical object from these varied simulated retrievals, the model learns to disentangle and selectively utilize shared geometric and appearance features.

**2D Data Mode: Held-out View Prediction.**   When training with 2D samples, the objective is to predict the single held-out image $I_{\text{target}}$ based on the text prompt $p$ and tokens from the $K$ conditioning retrieved images $\mathcal{I}_{\text{ret}}$. In this scenario, as only a single target view is generated, the U-Net's self-attention layers inherently function as standard 2D self-attention, operating within that single view's features. The text prompt $p$ and the tokens from the retrieved images provide conditioning via $f_{\text{txt}}$ and $f_{\text{ret}}$ respectively.

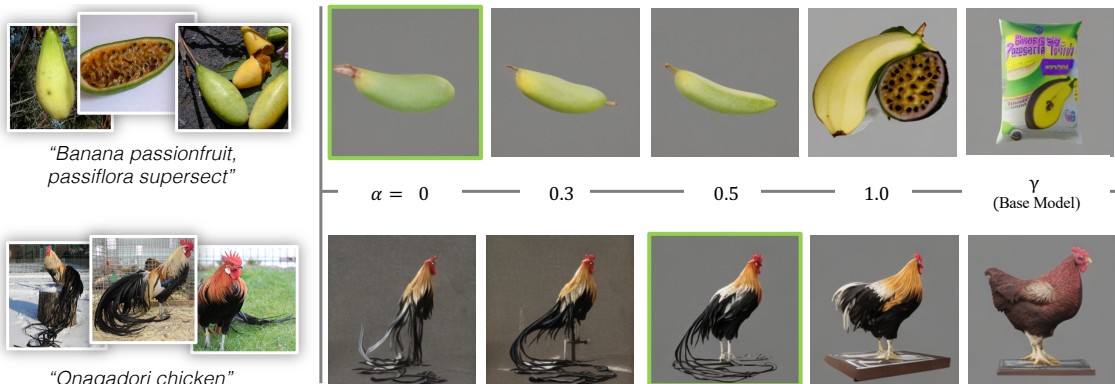

Figure 4: **Illustration of Prior-Guided Attention.** The base model's activations are leveraged proportionally to its prior knowledge of the object, controlled by the parameter $\alpha$. The results using prior-guided attention are marked with the green stroke.

Crucially, no explicit camera pose information is provided. This is a deliberate design choice: withholding pose information forces the model to infer 3D spatial relationships directly from unstructured 2D data. This data may often depict shared geometry or textures from different perspectives, which are noisy or view-incomplete. Rather than introducing noise, this 2D training acts as a structured regularizer, expanding the model's appearance generalization without overwriting the strong structural priors acquired during 3D training.

### 3.5 Inference Process

At inference time, given an input text prompt $p$, we first retrieve the top $K$ relevant 2D images $\mathcal{I}_{\text{ret}}$ from a 2D database using the BM25-based text similarity approach. To improve relevance, we compute prompt-caption similarity and discard images below a threshold, yielding $K' \leq K$ images. If no images pass the threshold, we disable retrieval-attention and fall back to the base model. These $K$ images are then encoded into visual tokens as detailed in Section 3.2. Our trained multiview diffusion model then generates $N$ consistent views conditioned on the text prompt $p$ and the set of retrieved tokens, utilizing specified camera poses for the target views.

**Prior-guided attention.** We introduce an adaptive fusion coefficient $\alpha$ that dynamically balances the influence of the model's prior knowledge and the retrieved signals, based on how OOD a prompt is. Diffusion models learn to approximate the score function $\nabla_x \log p(x|y)$, the gradient of the log data density which by definition points toward higher-probability regions of the data distribution (Song & Ermon, 2020). Thus, if a concept is in-domain, the base model's score will guide denoising toward an accurate reconstruction, whereas for OOD concepts the reconstruction will deviate.

To estimate $\alpha$ during inference, without ground-truth multiview, we first perform a short forward pass using only the base model's text-based attention $f_{\text{txt}}$ (retrieval module disabled) for 10 DDIM steps, generating an initial output. We then measure its similarity to retrieved images using DINOv2 similarity, which serves as a proxy for the base model's confidence in capturing the concept: high similarity indicates in-domain, so $\alpha$ favors $f_{\text{txt}}$; low similarity suggests OOD, shifting weight toward the retrieval-based attention $f_{\text{ret}}$. The two sources are fused adaptively as: $f = \alpha \cdot f_{\text{txt}} + (\lambda' - \alpha) \cdot f_{\text{ret}}$, for a hyperparameter $\lambda'$, replacing the $f$ calculation used in training. To handle extreme OOD concepts where retrieved images may be entirely irrelevant, a gating mechanism filters out retrievals below a minimum BM25 threshold. If no images pass, the model smoothly falls back to the base model prior. While $\alpha$ operates as an efficient 2D proxy, empirical sensitivity analysis confirms it is highly robust around its optimal operating point (see Appendix A.2). The model is then run with the retrieval module enabled to generate final outputs. Fig. 4 illustrates its effect. Crucially, for in-domain concepts, this dynamic fusion naturally down-weights external retrieval, effectively shielding in-domain performance from noisy retrievals.

Table 1: **Quantitative evaluation on OOD/rare concepts.** The models' performance is assessed on four orthogonal views. See Sec. 4 for further details.

| Method | 4-Views | | | | | Re-rendered (3D Reconstruction) | | | | |
|---|---|---|---|---|---|---|---|---|---|---|
| | CLIP ↑ | DINOv2 ↑ | IR ↑ | FID ↓ | IS ↑ | CLIP ↑ | DINOv2 ↑ | IR ↑ | FID ↓ | IS ↑ |
| **Text-to-3D** | | | | | | | | | | |
| MVDream | 66.47 | 33.12 | 58.01 | 76.71 | 10.62 | 70.83 | 28.66 | 58.98 | 96.29 | 11.39 |
| MV-Adapter (TX) | 66.48 | 28.53 | 58.42 | 84.28 | 9.55 | 71.33 | 24.30 | 56.14 | 106.66 | 11.23 |
| SPAD | 65.23 | 19.39 | 48.54 | 167.49 | 9.18 | 64.46 | 12.29 | 43.80 | 176.66 | 8.90 |
| TRELLIS (TX) | 67.96 | 21.11 | 51.01 | 160.93 | 6.90 | 67.16 | 16.87 | 51.82 | 154.43 | 8.09 |
| **Image-to-3D** | | | | | | | | | | |
| ImageDream-P | 69.20 | 45.01 | 65.64 | 68.40 | 12.11 | 70.44 | 32.77 | 60.17 | 103.24 | 12.84 |
| ImageDream-L | 67.55 | 39.48 | 63.93 | 84.69 | 9.45 | 70.16 | 29.60 | 58.66 | 120.37 | 10.42 |
| MV-Adapter (IM) | 69.74 | 49.14 | **71.05** | 72.71 | 12.88 | 71.53 | 35.25 | 60.36 | 107.95 | 12.64 |
| Era3D | 69.13 | 42.41 | 64.42 | 92.68 | **15.26** | 71.00 | 35.65 | 60.81 | 93.97 | **14.45** |
| TRELLIS (IM) | 70.31 | 35.24 | 59.32 | 167.61 | 11.38 | 67.86 | 24.43 | 52.23 | 146.82 | 10.72 |
| **3D Personalization** | | | | | | | | | | |
| MVDreamBooth | 66.14 | 36.22 | 55.09 | 82.73 | 11.55 | 68.38 | 27.91 | 54.33 | 107.07 | 11.92 |
| **MV-RAG (Ours)** | **71.77** | **50.19** | 67.41 | **54.79** | 13.20 | **74.28** | **39.61** | **66.59** | **80.54** | 12.33 |

# 4 Experiments

We evaluate our approach to state-of-the-art baselines on both OOD/rare and in-domain concepts.

**Benchmarks** As current benchmarks lack OOD/rare concept coverage, we curated 196 examples from Wikipedia Commons (Wikimedia Commons, 2025) (not used in training). Each example consists of a text prompt and multiple 2D retrieved images of the same concept. Importantly, texts were chosen to be far from any text (or concepts) seen during training. We call this evaluation set "OOD-Eval". See Appendix Sec.A.3.6 for additional details and examples. We also consider in-distribution objects, demonstrating that our success in OOD concepts is not compensated by worse in-domain results. To this end we consider a curated set of 50 in-domain objects from Objaverse-XL (Deitke et al., 2023a). For retrieval, we consider 2D images from the LAION-400M dataset (Schuhmann et al., 2021). For each text, we retrieve four 2D images. Unlike for OOD-Eval, we also have corresponding ground truth multiview images. We call this evaluation set "IND-Eval".

**Baselines** We compare MV-RAG against three categories of state-of-the-art methods. (1) *Text-to-multiview* generation: MVDream (Shi et al., 2023b), MV-Adapter (Huang et al., 2024) (text-conditioned), SPAD (Kant et al., 2024), and TRELLIS (Xiang et al., 2025) (text-conditioned). (2) *Image-to-multiview* generation applied to the retrieved views: ImageDream (Wang & Shi, 2023), MV-Adapter (Huang et al., 2024) (image-conditioned), Era3D (Li et al., 2024), and TRELLIS (Xiang et al., 2025) (image-conditioned). (3) *3D personalization*: we adopt MVDream's optimization-based personalization approach (Shi et al., 2023b; Ruiz et al., 2023), applied to all $k$ retrieved views. Unlike prior work, MV-RAG leverages multiple ($k = 4$ in our experiments) retrieved in-the-wild images that may differ significantly in pose, setting, and object identity. To the best of our knowledge, MV-RAG is the first framework to effectively use such diverse multi-image inputs for object multiview generation. To ensure a fair comparison, we adapt baselines accordingly: for image-to-multiview methods, we prompt each model separately with every retrieved condition image and report the best-scoring output across them. This setup provides baselines with access to the same retrieval set while respecting their single-image conditioning design. Further details are provided in Appendix Sec.A.3.4.

## 4.1 Quantitative Evaluation

**Metrics** We assess generation quality with Inception Score (IS) (Barratt & Sharma, 2018) and FID (Heusel et al., 2018) on the output poses. To assess alignment to the input text, a natural choice would be to

Table 2: **Quantitative evaluation on in-domain concepts.** Our method maintains competitive performance with baselines on standard in-distribution objects, demonstrating that our OOD capabilities do not come at the cost of in-domain quality.

| Model | PSNR ↑ | SSIM ↑ | LPIPS ↓ | CLIP ↑ | SigLIP ↑ |
|---|---|---|---|---|---|
| **Text-to-3D** | | | | | |
| MVDream | **16.95** | 0.717 | 0.363 | 64.25 | 34.81 |
| MV-Adapter (TX) | 15.37 | 0.632 | 0.459 | 59.62 | 30.18 |
| SPAD | 8.34 | 0.619 | 0.468 | 61.32 | 29.12 |
| TRELLIS (TX) | 16.53 | **0.743** | **0.327** | 60.67 | 30.98 |
| **Image-to-3D** | | | | | |
| ImageDream-P | 15.50 | 0.728 | 0.400 | 60.89 | 31.67 |
| ImageDream-L | 15.64 | 0.732 | 0.393 | 61.72 | 32.52 |
| MV-Adapter (IM) | 15.24 | 0.646 | 0.448 | 61.46 | 32.00 |
| Era3D | 12.44 | 0.722 | 0.378 | 58.79 | 29.94 |
| TRELLIS (IM) | 16.02 | 0.741 | 0.378 | 55.39 | 26.72 |
| **3D Personalization** | | | | | |
| MVDreamBooth | 16.31 | 0.716 | 0.381 | 61.68 | 32.14 |
| MV-RAG (Ours) | 16.63 | 0.730 | 0.362 | **64.48** | **35.34** |

consider the CLIP (Radford et al., 2021) similarity between the input text and the output multiview images produced by the model. However, we found that CLIP (specifically in image-text similarity) is unable to score rare/OOD concepts well, often assigning a low score for such text-image pairs. See Appendix A.2.1 and Fig. 9 for further discussion and illustration. This is also demonstrated in (Zhu et al.). As such, we evaluate image-image similarity between the generated views and held-out ground truth retrieved examples from our evaluation benchmark. Specifically, we compute the average similarity using CLIP and DINOv2 (Oquab et al., 2023). Additionally, we employ an *Instance Retrieval (IR)* model (Shao & Cui, 2022) specifically trained to embed images of the same object instance close together in feature space, making it a more suitable choice for assessing entity-level visual alignment.

To evaluate 3D consistency, we adopt the procedure of (Wang & Shi, 2023), measuring how well a *3D reconstruction* model aligns with our generated views. Specifically, we use 4 generated views and train a *feed-forward 3D reconstruction* model (Tang et al., 2024a) on these "training views". We then render and evaluate 18 novel views (Re-rendered in Tab. 1), whose fidelity and alignment with the training views are assessed using the metrics described above. Inconsistent multiview generations are expected to degrade reconstruction quality, leading to lower fidelity and alignment scores. Since geometric inconsistency is a primary failure mode for diffusion models on OOD concepts, this directly measures our method's ability to generate coherent 3D objects for rare prompts. Further details are in Appendix Sec.A.4.

**User study.** We complement our quantitative metrics with a user study on OOD-Eval prompts. Participants rated sets of four generated view from our model and baselines on a 1-5 scale for *Realism* (Q1), *Text Alignment* (Q2), and *3D Consistency* (Q3). See Appendix A.4 for full details.

Table 3: **User study.** Mean Opinion Scores (MOS; 1 = low, 5 = high) for (Q1) Realism, (Q2) Alignment, and (Q3) 3D Consistency.

| | Q1 ↑ | Q2 ↑ | Q3 ↑ |
|---|---|---|---|
| MVDream | 1.96 | 1.85 | 3.24 |
| ImageDream-P | 2.25 | 2.60 | 3.03 |
| MV-RAG (Ours) | **4.12** | **4.44** | **4.44** |

**Evaluation on OOD/rare concepts** As shown in Tab.1, MV-RAG achieves strong performance across both evaluation modes. In the *4-views* setting, it outperforms all baselines on CLIP, DINO, and FID, while ranking second on IR (behind MV-Adapter (IM)) and IS (behind Era3D). In the more challenging *rerendered* setting which also reflects 3D consistency, MV-RAG leads on CLIP, DINO, IR, and FID, with Era3D attaining a higher IS. Notably, MVDream and ImageDream, which share similar architectures but lack retrieval, consistently underperform across metrics. The user study results in Tab.3 further corroborate

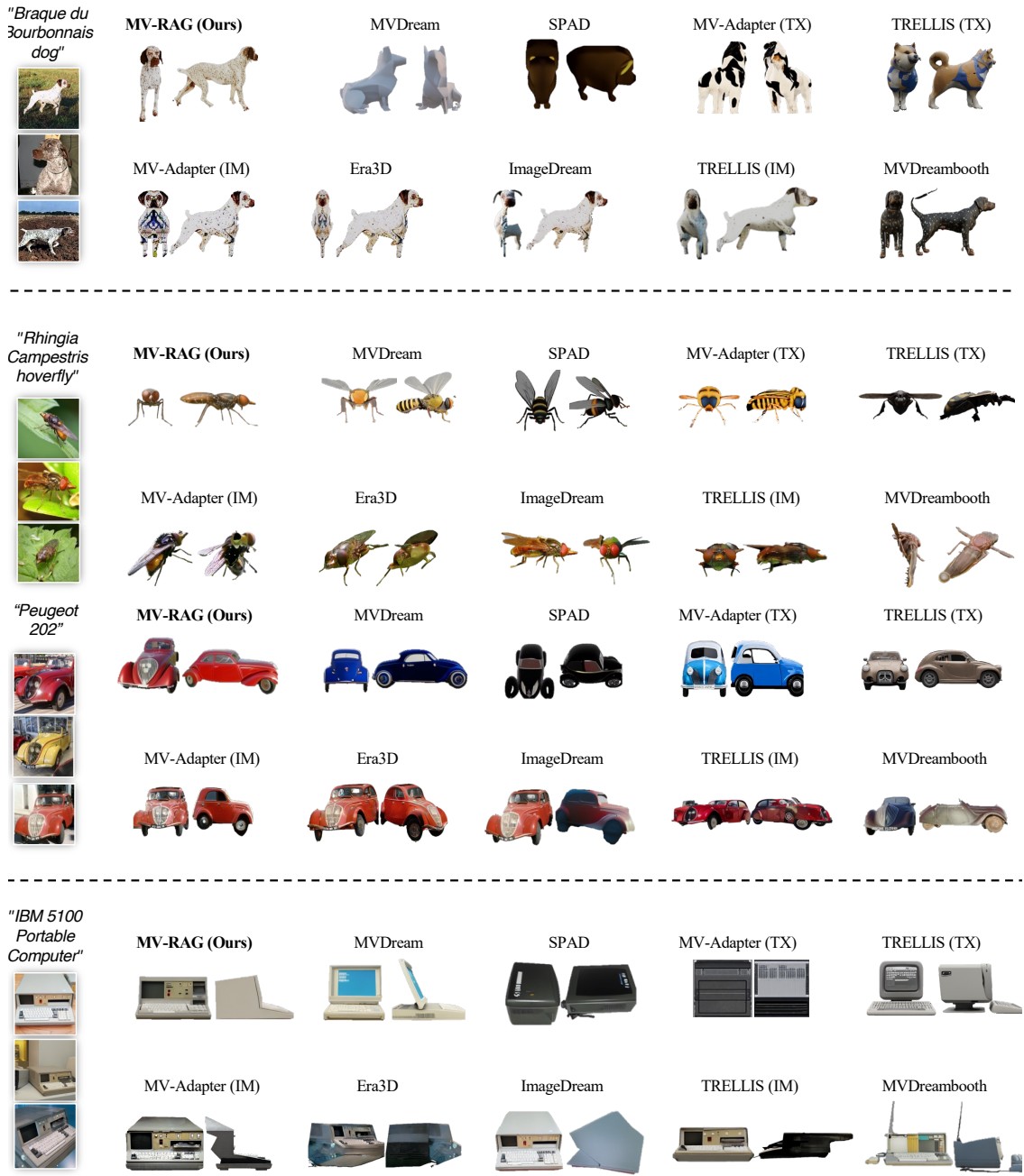

Figure 5: **Qualitative evaluation.** Text-to-3D models fail to generate unseen (OOD) concepts, while image-to-3D models fail to reconstruct correct 3D structure from a single view. Existing personalization methods cannot effectively leverage the diversity of retrieved images. For each object we present MV-RAG results and text-to-3D methods in the top row, image(s)-to-3D methods in bottom row. **Zooming in is suggested.** Additional full-scale evaluation is presented in the supplementary materials.

MV-RAG's advantage, showing clear gains in realism, text alignment, and 3D consistency. Our model's superior 3D consistency on OOD concepts, shown in the "Re-rendered" evaluation, is a direct result of our training. Unlike baselines whose internal priors are insufficient for OOD prompts, MV-RAG is explicitly taught in its 3D training mode to extract partial geometric cues from noisy, unposed views.

**Evaluation on in-domain concepts** We evaluate MV-RAG against all methods on the IND-Eval benchmark, which contains objects from the Objaverse (Deitke et al., 2023a) dataset that is used for training

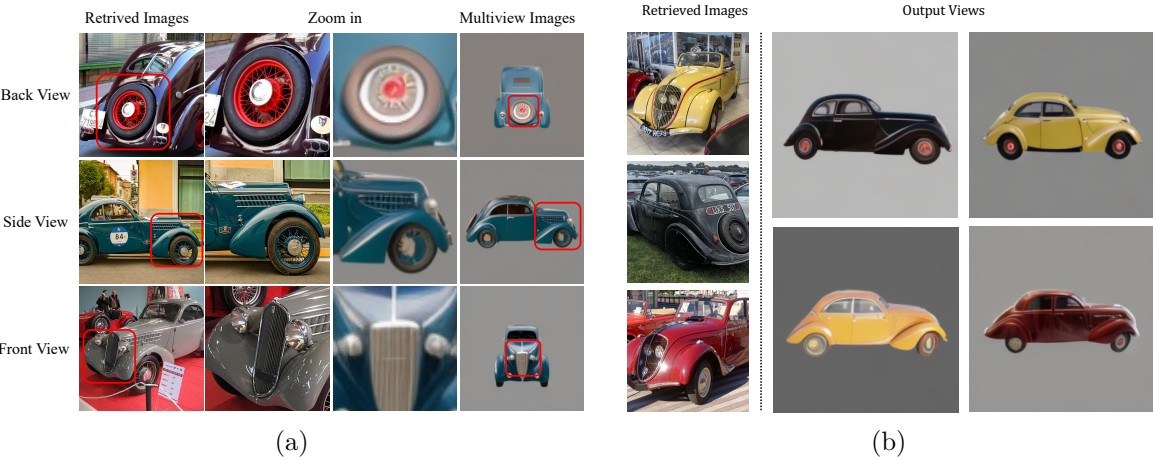

(a)                      (b)

Figure 6: (a) **Utility.** On the LHS, we show retrieved views. The middle columns are zoom-ins for aspects used in generation, and the RHS shows back view (top), side view (middle), and front view (bottom). (b) **Diversity.** For the prompt "Peugeot 202", the LHS shows retrieved views, and the RHS shows a single-view output (using the same pose) for four different seeds.

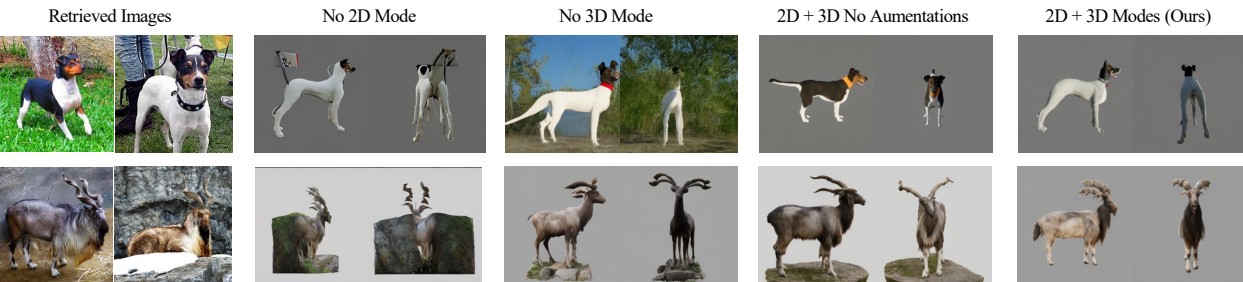

Figure 7: **Hybrid training ablations.** Output of *"Ratonero Bodeguero Andaluz dog"* (top) and *"Markhor goat"* (bottom), comparing models trained with (RHS) and without our 2D/3D schemes and augmentations.

in all baselines. Reconstruction quality is measured using PSNR, SSIM, and LPIPS with respect to the ground-truth views in IND-Eval, while text-image alignment is assessed via CLIP and SigLIP (Zhai et al., 2023) similarity between the generated outputs and the input prompt. As shown in Tab. 2, MV-RAG achieves results that are on par with, or slightly surpass those of the baselines.

## 4.2 Qualitative Evaluation

Fig. 5 compares MV-RAG to baselines (see Appendix Figs. 17, 18 for more). Text-only models often fail on OOD objects, lacking visual priors and producing incorrect geometry. Single-reference image-to-3D methods are constrained by their single viewpoint: although they achieve high similarity to the input image, they cannot infer the true 3D structure for OOD objects, leading to smudging and artifacts in unobserved regions. Even multi-reference approaches like MVDreamBooth struggle to integrate diverse cues effectively, resulting in inconsistent colors, textures, and geometry. MV-RAG overcomes these limitations by leveraging multiple unposed images from a large 2D corpus, providing complementary viewpoints that enrich generation with relevant visual cues. Crucially, these improvements stem from visual identity grounding and fine-grained texture cues rather than explicit geometry cues, compensating for the base model's weak semantic understanding of rare concepts; the base model contributes the coarse structural layout, while retrieval injects specific fine-grained visual features. Our framework isolates view-invariant attributes such as object texture while disentangling nuisance factors like illumination, occlusion, and background, producing diverse and accurate multiview outputs (Fig. 5).

Table 4: Quantitative ablation study for the hybrid training scheme, calculated on re-rendered views.

| Model | CLIP | DINO | IR | FID | IS |
|---|---|---|---|---|---|
| No 2D-mode | 72.995 | 39.100 | 66.196 | 82.046 | **13.065** |
| No 3D-mode | 73.435 | 37.973 | 65.557 | 84.614 | 12.320 |
| MV-RAG | **74.278** | **39.608** | **66.588** | **80.540** | 12.325 |

**Diversity and Utility.** Unlike image-prompted methods, MV-RAG can produce diverse outputs for the same text prompt by varying the random seed, as illustrated in Fig. 6(a), where the model extracts and blends distinct components (e.g., a spare wheel or car grille) from different retrieved views. Moreover, Fig. 6(b) highlights MV-RAG's ability to leverage multiple retrieved views: the model combines information from different source views to generate consistent target views.

## 4.3 Ablation Studies

**Hybrid training** Fig. 7 presents a qualitative ablation of our 2D mode, 3D mode, and augmentations. The 2D mode is essential for handling real-world complexities such as occlusions, background clutter, non-canonical poses, and multiple objects. Without it, the model fails to separate object features from environmental context, causing artifacts (e.g., a floating leash on a dog, a goat merged with a rock) that are 'baked' into the 3D geometry and degrade novel-view renderings. The 3D mode, by contrast, enforces multi-view consistency and geometric accuracy, ensuring that visual features are correctly distributed across views; without it, shapes are distorted and background inconsistencies arise (e.g., tails or horns missing or deformed).

These complementary roles are quantitatively reflected in the re-rendered evaluation in Table 4. Removing the 2D mode lowers semantic alignment and visual quality, while removing the 3D mode degrades geometric accuracy and multi-view consistency. Removing augmentations still allows in-the-wild settings through the 2D mode but reduces robustness to high variance in retrievals, yielding incorrect 3D structures.

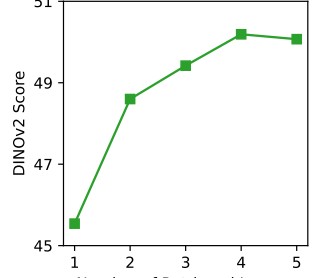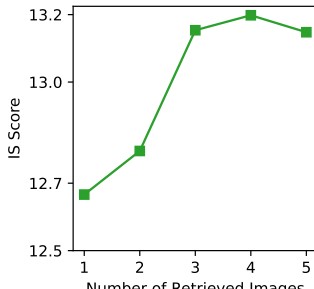

Figure 8: Effect of the number of retrieved images on alignment and fidelity.

**Number of Retrieved Images** Fig. 8 shows the effect of the number of retrieved images on alignment and fidelity. Using four views yields the best performance, as multiple exemplars provide complementary cues about geometry and texture, helping the model capture 3D structure under varying conditions. Beyond four views, gains saturate, suggesting redundancy rather than additional useful information.

**Retrieval method.** We compare BM25 with semantic dense retrievers, CLIP (Radford et al., 2021) and SigLIP (Zhai et al., 2023), on a combined OOD-Eval and MS-COCO (Lin et al., 2014) corpus. Table 5 reports retrieval performance using OOD-Eval prompts, where the goal is to retrieve the correct image-text pair. We evaluate CLIP using text-to-image (TX-IM) and text-to-text (TX-TX) similarity, SigLIP with TX-IM similarity, and BM25 as a lexical baseline. Dense semantic retrievers often underperform in the OOD setting due to limited exposure or weak grounding: as shown in Fig. 9 and Table 6, CLIP frequently assigns high scores to incorrect or overly generic matches, failing to distinguish rare objects from broad categories. By contrast, BM25 is more robust, relying on direct keyword overlap rather than learned semantic priors.

## 5    Conclusion

We introduce a retrieval-augmented multiview diffusion model for text-to-3D generation. By conditioning on relevant 2D images from a large database, our method produces consistent and accurate multiview outputs, particularly for out-of-domain (OOD) or rare concepts where prior methods struggle. A hybrid training scheme integrates structured multiview data with diverse 2D collections, using augmented conditioning views and a held-out view prediction objective to bridge the gap between posed 3D assets and unposed real-world images. To further robustify generation, we incorporate a prior-guided fusion mechanism that dynamically balances the model's internal knowledge with external retrieval signals. To evaluate challenging cases, we introduce a new OOD benchmark. Experiments show that our approach substantially improves 3D consistency, photorealism, and text alignment for OOD concepts while maintaining strong performance on standard benchmarks.

## 6    Broader Impact

MV-RAG significantly advances text-to-3D synthesis, particularly by improving the accuracy and geometric consistency of out-of-domain and rare concepts. However, because the framework relies on retrieving images from web-scale, in-the-wild 2D databases, it risks inheriting and propagating the inherent social, cultural, or demographic biases present in those datasets into the synthesized 3D assets. Furthermore, the framework's powerful capability to accurately generate rare or highly specific concepts introduces risks related to copyright and intellectual property infringement (e.g., replicating proprietary designs or characters), as well as the potential generation of deceptive or malicious 3D assets. To mitigate these concerns, future applications of this technology should employ strict content filtering on the text prompt and retrieval corpus, prioritize the use of ethically sourced databases, and implement digital watermarking or provenance tracking for the generated 3D outputs.

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

# A Appendix

## A.1 Additional Qualitative Evaluation

We provide further qualitative results to complement Fig. 5 from the main paper. Comparative examples are shown in Fig. 17 and Fig. 18, while additional outputs generated by our method are presented in Fig. 15 and Fig. 16.

## A.2 Additional Experiments

### A.2.1 OOD for Representation Models

Figure 9 highlights three representative failure cases of using CLIP text-image similarity as an evaluation metric for out-of-distribution (OOD) objects. In each case, MVDream receives a higher CLIP score than both our model and even the ground truth image, despite generating outputs that are visually or semantically incorrect. We hypothesize that this stems from CLIP's limited prior knowledge of rare concepts and the fact that models like MVDream are optimized to align with CLIP-based features, potentially leading to overfitting to incorrect semantic associations. Further, as shown in Table 6, we find that for rare concepts, CLIP assigns nearly identical similarity scores to both detailed object names and their coarse class labels. This suggests that CLIP does not treat the additional semantic information in rare object names as meaningful, highlighting a lack of conceptual grounding for these OOD categories. In contrast, for in-domain objects, CLIP shows much stronger separation between specific and generic labels, reinforcing its limitations in recognizing and evaluating uncommon or unseen concepts.

These observations underscore the limitations of using CLIP for OOD evaluation and motivate our decision to adopt image-image similarity metrics instead, which more reliably reflect visual fidelity. To this end, we employ CLIP (Radford et al., 2021), DINOv2 (Oquab et al., 2023), and an Instance Retrieval (IR) model (Shao & Cui, 2022) fine-tuned from CLIP to better align visual object instances.

Figure 9: **Limitations of CLIP text-image similarity for evaluating OOD objects.** Each row shows an example from our OOD-Eval benchmark: the ground truth (GT) image, the output from MVDream, and the output from our model. Below each image is its CLIP similarity score. MVDream receives a higher score than both our model and the GT image, despite producing less faithful generations.

### A.2.2 Retrieved images with distinct appearances.

We conducted a controlled experiment on OOD entities to test the effect of retrieval variance. We compared three strategies: (a) low variance (highly similar images), (b) moderate variance (our default top-k retrieval), and (c) high variance (deliberately diverse images). We observed that low variance yields accurate but less diverse generations, sometimes propagating instance-specific bias (see Sec. A.5 and Fig. 14)(a). Our default moderate variance offers the best balance of quality and diversity. High variance can sometimes challenge the model's ability to recover a consistent 3D structure, though the results remain superior to baselines given the same challenging inputs.

Table 5: Comparison of retrieval approaches for out-of-domain retrieval.

| Method | Precision@5 ↑ |
|---|---|
| CLIP (TX-IM) | 0.5366 |
| CLIP (TX-TX) | 0.7306 |
| SigLIP (TX-IM) | 0.7889 |
| BM25 (TX-TX) | **0.8522** |

### A.2.3 Noisy retrievals (n < k relevant images).

Table 6: **CLIP similarity between retrieved images and concept labels.** We compute similarity to the GT retrieved images with the object name against the class label (e.g., "dog", "car") and report average, max, and their absolute difference. OOD examples show minimal semantic separation.

| Domain | Text | Avg | Max |
|--------|------|-----|-----|
| OOD | Bucovina Shepherd Dog | 63.59 | 67.16 |
| | Dog | 63.41 | 67.45 |
| | *Abs. Diff* | *0.18* | *0.29* |
| | BMW 319 automobile car | 66.33 | 71.11 |
| | Car | 65.14 | 71.56 |
| | *Abs. Diff* | *1.19* | *0.45* |
| In-Domain | Airedale Terrier dog | 89.54 | 96.58 |
| | Dog | 64.23 | 69.99 |
| | *Abs. Diff* | *25.31* | *26.59* |
| | American Hairless Terrier dog | 83.08 | 92.79 |
| | Dog | 64.59 | 70.11 |
| | *Abs. Diff* | *18.49* | *22.68* |

Our method is robust to retrieval noise. As described in Section 3.5, we employ a gating mechanism that filters out irrelevant images based on a similarity threshold. Specifically, for each retrieved image, if its BM25 score was lower than a given threshold (9.36), we do not consider it, and use the rest. Our performance in this case, therefore, translates to only using $K$ (=1,2,3, or 4) relevant retrieved images. This ablation is shown in Fig. 8, demonstrating that as $K$ increases, performance is improved.

### A.2.4 Incorrect alpha scores in prior-guided attention.

The adaptive score $\alpha$ is critical. If $\alpha$ is underestimated for an OOD object, the model relies too heavily on its weak prior, and the output resembles a degraded baseline generation. Conversely, if $\alpha$ is overestimated, the model may overly trust the base model's prior even when it is flawed, potentially inheriting 3D structural errors (e.g., a floating tail), as discussed in Sec. A.5 and Fig. 14(c)).

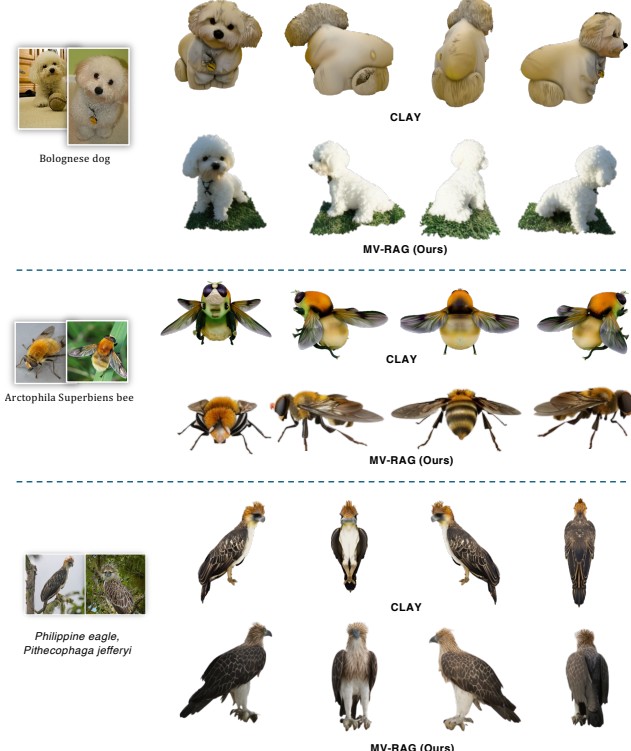

Figure 10: Qualitative Comparison against CLAY.

### A.2.5 Challenges of Multiview RAG In Comparison to Single-View RAG

While our approach builds upon the general idea of RAG, adapting it to multi-view generation introduces unique and substantial challenges that go beyond single-image RAG settings:

Table 7: User study comparing the multiview outputs of MV-RAG against the CLAY (Rodin) demo. Our method was rated significantly higher in realism, alignment, and 3D consistency.

| Method | Q1: Realism ↑ | Q2: Alignment ↑ | Q3: 3D Consistency ↑ |
|---|---|---|---|
| CLAY (Rodin) | 2.20 | 2.40 | 3.95 |
| **MV-RAG (Ours)** | **4.60** | **4.60** | **4.63** |

1. **Cross-view consistency from partial retrievals.** Single-image RAG only requires synthesizing a single image from the retrieved content. In contrast, multi-view RAG must ensure geometric consistency across several generated views. For example, if the model leverages a retrieved image that reveals the bonnet of a car for the front view, it must synthesize a side view that is consistent with that bonnet, even if the side view is not present or visible in any of the retrieved images. This kind of spatial reasoning is not required in single-view generation.

2. **View-specific and consistent use of different retrieved images.** Our method learns to draw on different retrieved images for different generated views, depending on which object components are visible. For instance, in Fig. 6(a), the model extracts the spare wheel, car grille, and front wheel from separate retrieved images and integrates them into different output views (back, front, and side). To do this effectively, the model must reason about 3D structure and spatial layout, a requirement absent from single-image RAG.

**How our method addresses (1) and (2)?** To address these issues, we designed a framework that can (i) selectively attend to relevant parts of the retrieved views for each target view and (ii) enforce consistency across generated images through shared conditioning and a hybrid training scheme. This is especially challenging because the retrieved views are in-the-wild and unposed, and no dataset provides direct supervision for multi-view consistency across retrieved parts.

### A.2.6 Comparison with CLAY (Rodin Demo)

We also performed a qualitative comparison with CLAY (Zhang et al., 2024a), for which a public model was not available at the time of our evaluation. We used their publicly accessible online demo, Rodin[1], to generate multiview images for several prompts from our OOD-Eval set. We then conducted an additional user study following the protocol described above. Participants were shown randomly ordered sets of multiview images from Rodin and our method and were asked to rate them on a scale of 1-5 across three attributes. As summarized in Table 7, the outputs from MV-RAG were strongly preferred by users across all criteria. See Figure. 10 for comparison visualization.

### A.2.7 Limitation
### of Scaling Data and Parameters for OOD

A natural question is whether out-of-distribution (OOD) challenges can be resolved simply by using a stronger model. To investigate this, we evaluate a strong text-to-image model (FLUX.1 (Labs et al., 2025)) on several OOD prompts from our OOD-Eval benchmark (See Figure 11).

*Arctophila superbiens bee*

*Banana passionfruit, passiflora supersect*

*Braque du Bourbonnais dog*

*BMW 319 automobile car*

Prompt      T2I Output      Ground Truth

Figure 11: Large scale text-to-image model (FLUX.1) outputs on prompts from OOD-Eval.

[1] https://hyper3d.ai/

Target Multiview — Augmented Views

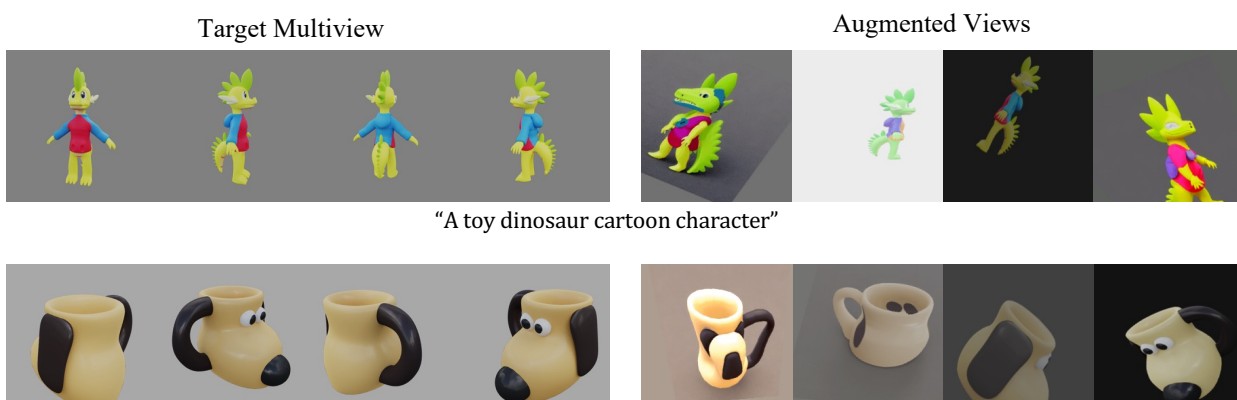

"A toy dinosaur cartoon character"

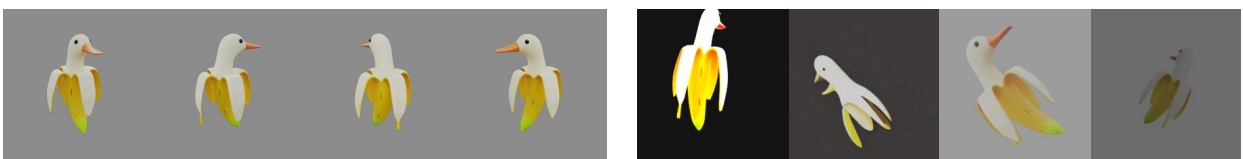

"Dog-shaped coffee mug with white and brown colors, and features like eyes and nose"

"A banana-shaped toy bird"

Figure 12: **Illustration of the augmented views in 3D mode.** Examples from the training set in 3D mode. For each target multiview instance, retrievals are simulated using photometric augmentations (crop, color jitter, perspective transform, etc.) and neural augmentations via an image-variation model.

Despite its scale and capacity, it frequently fails on these prompts, producing inaccurate structures, missing characteristic details, or hallucinating unrelated content.

This illustrates a fundamental property of OOD generalization. We argue that it cannot be solved solely by increasing model capacity or training data. OODness arises from distributional gaps rather than a fixed set of missing visual features understanding, and there will always exist concepts that fall outside a model's training distribution. Newly emergent or rare objects (e.g., a "Labubu doll") exemplify this challenge. Retrieval-based approaches such as MV-RAG address this limitation by conditioning generation on real visual exemplars, providing adaptability and robustness that scaling alone cannot achieve.

Table 8: Average inference time for each model.

| Model | Time (seconds) |
|---|---|
| MVDream | 1.081 |
| ImageDream | 1.470 |
| MV-Adapter | 6.562 |
| SPAD | 10.250 |
| Trellis | 12.098 |
| Era3D | 12.640 |
| MVDreambooth | 170.411 |
| **MV-RAG** | 6.296 |

### A.2.8 Inference time

Table 8 reports the average inference time for all methods. For MV-RAG, the average generation time is 6.26 seconds, with an additional 0.04 seconds spent on retrieval, resulting in a total inference time of 6.30 seconds. This confirms that the retrieval overhead is negligible and that the introduced adapters incur only a minimal computational cost, preserving the overall efficiency of the model.

### A.3  Training, Inference and Implementation Details

### A.3.1  Data Preparation

For 3D mode training, we utilize multi-view synthetic renderings from the public Objaverse dataset (Qiu et al., 2024; Deitke et al., 2023b) along with the associated camera parameters. We randomly sampled a subset of 90K objects. For each object, we select four orthogonal views with elevation angles in the range $[-5°, 30°]$ for supervision. Additionally, we sample 2-3 random views to simulate retrieval, as detailed below. Objaverse contains a wide variety of objects, including both high-fidelity, photorealistic assets and low-textured, abstract ones. To improve the model's robustness to real-world, non-synthetic data, we apply an aesthetic-based filtering criterion. This criterion incorporates color diversity, texture complexity, and multi-view consistency, which then results in about 65K objects. Following the preprocessing protocol of MVDream, we resize all rendered images to $256 \times 256$ pixels and replace empty backgrounds with a random gray color. Camera poses are normalized onto a unit sphere by removing translational components.

To simulate retrieval images, we apply a series of augmentations to the additional rendered views. These include perspective distortion, random rotations, resized cropping, and color jitter. To further enhance realism, we employ an image-variation model (Ye et al., 2023; Rombach et al., 2022) to generate semantically and visually diverse variants of the same object. In total, we obtain four simulated retrieval images per object. We present an illustration for the augmentations in Fig. 12. For 2D mode training, we use the ImageNet21K dataset (Ridnik et al., 2021), which comprises over 21K semantic classes with multiple images per class. To improve the visual coherence within classes, we use a large language model (GPT-4o) to filter and retain only visually unified categories (e.g., *carpet shark*, *toilet bowl*) and exclude abstract or overly broad classes (e.g., *human*, *cycling*). This results in a curated subset of 516 visually consistent categories. For each selected class, we sample one target image for supervision and four additional images from the same class to serve as retrieved images.

### A.3.2  Retrieval Process

We evaluate multiple retrieval strategies based on both image-text and text-text similarity. For embedding-based retrieval with CLIP (Radford et al., 2021) and SigLIP (Zhai et al., 2023), we build an index using the FAISS library (Douze et al., 2025), which supports efficient approximate nearest-neighbor search in high-dimensional spaces. We additionally employ Pyserini (Trotman et al., 2014; Robertson et al., 1994; Lin et al., 2021) for text-based retrieval using the BM25 ranking function. This approach is a highly optimized and scalable toolkit designed for large-scale retrieval tasks, capable of indexing millions of documents while providing fast query responses. Its retrieval time is typically sub-linear with respect to corpus size due to inverted index structures, enabling near real-time search performance with minimal computational overhead, as demonstrated in large-scale search engine systems.

### A.3.3  Training

We fine-tune our model using the AdamW optimizer(Loshchilov & Hutter, 2019) with a learning rate of $5 \times 10^{-6}$ and a batch size of 24 for approximately 11,000 steps. Training is performed in an alternating scheme between 2D and 3D modes, allocating an equal number of steps to each mode. As in MVDream, we append ", 3d asset" to the text prompt during 3D mode to help the model distinguish between the two training regimes. The model is initialized from the Stable Diffusion 2.1-based MVDream checkpoint, which remains frozen throughout training. The adapter modules are initialized from the ImageDream checkpoint. We fine-tune both the retrieval-attention modules and the Resampler. For the image encoder, we use OpenCLIP ViT-H/14, which is kept frozen during training. The training was done on a single NVIDIA A100 GPU, with a total training time of approximately 3 hours.

### A.3.4  Baselines

For all baselines we use the official implementations and publicly available pretrained checkpoints provided by the respective authors, with the exception of MVDreamBooth, for which training code is not released. For each baseline, we generate 4 views using fixed orthogonal camera angles and elevations, employing the

IND-Eval                                                     OOD-Eval

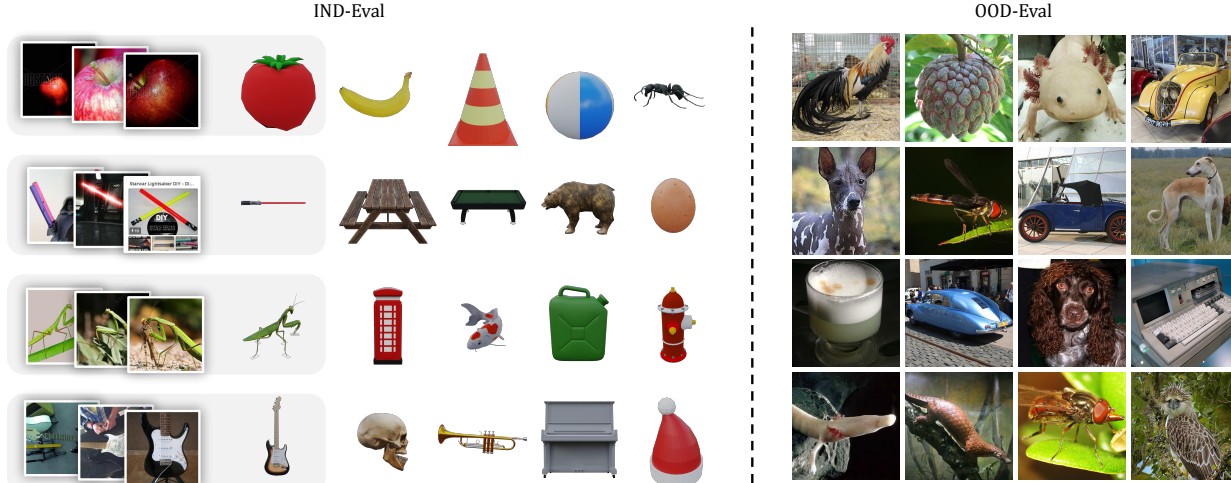

Figure 13: **Evaluation Benchmarks Overview. OOD-Eval:** Our out-of-distribution benchmark includes 2D images of both rare and well-known objects, featuring a diverse set of categories such as animals, vehicles, insects, foods, and everyday items. **IND-Eval:** The in-domain benchmark focuses on common, everyday objects that are representative of standard training distributions.

DDIM sampler with 50 steps and a classifier-free guidance (CFG) scale of 5. For image-to-3D baselines, we preprocess the retrieved reference images by segmenting out the background using Grounded-SAM (Kirillov et al., 2023; Liu et al., 2023b; Ren et al., 2024). Among the reference images, we select the one that yields the highest multi-view consistency based on DINO score for evaluation. To ensure a fair comparison in image-image similarity metrics, we compare semantic features against the segmented ground-truth object views.

For the MVDreambooth baseline, we follow the method described in (Shi et al., 2023b), training a separate MVDreamBooth model for each instance in the OOD-Eval set. Each model is optimized for 600 steps. To preserve class identity, we apply a class-preservation loss using ImageNet class names (e.g., *dog*, *car*) when available, and default to the prompt when no corresponding class is derived.

### A.3.5  Re-rendering

To more thoroughly assess the 3D consistency and fidelity of the baselines on the OOD-Eval benchmark, we employ LGM (Tang et al., 2024a). Specifically, we reconstruct a 3D Gaussian representation using four output views generated by each model. From this reconstruction, we render 18 additional novel views sampled along a circular trajectory. These re-rendered views are then evaluated against the retrieved images using the image-image similarity metrics described earlier. We utilize the publicly available LGM implementation with its default configuration settings.

### A.3.6  Evaluation Dataset Construction

**Construction of OOD-Eval.**   We construct an evaluation benchmark, OOD-Eval, consisting of 196 objects. To ensure diversity and out-of-distribution coverage, we use a large language model (GPT-4o) to curate object names representing rare or unique concepts, as well as familiar objects that are absent from the training data. These include examples such as extinct or rare animal species, uncommon vehicles, and other atypical items. A visual preview of the benchmark is provided in Fig. 13.

For generating image captions, we leverage a vision-language model, specifically Qwen-VL (Bai et al., 2023), which provides high-quality textual descriptions of the images. These captions are used in the retrieval process (see Sec. 3.5).

| Set Comparison | BM25 ↑ | CLIP ↑ |
|---|---|---|
| OOD-Eval → LAION | 14.74 | 73.12 |
| LAION → LAION | 27.95 | 80.10 |

Table 9: Nearest-neighbor similarity between OOD-Eval prompts and LAION captions compared to a LAION in-distribution control.

**Construction of IND-Eval.** We constructed an in-domain evaluation set by selecting 50 well-known or everyday objects from the widely used Objaverse-XL dataset. For each object, we retrieve 4 reference images from the large-scale LAION-400M dataset (Schuhmann et al., 2021) using BM25-based text retrieval (see Sec. 3.5 in main paper). The retrieved images often exhibit significant visual or modality variation (e.g., artistic renderings or paintings of the object), as illustrated in Fig. 13.

**Quantifying Out-of-Distribution Distance.** To further assess how far OOD-Eval prompts lie from the training data distribution, we conduct a large-scale similarity analysis against LAION. We sample 100,000 LAION captions as a reference corpus. For each of the 196 OOD-Eval prompts, we compute its nearest neighbor in this corpus using both BM25 and CLIP similarity. As a control, we additionally sample 2,000 LAION captions (disjoint from the reference set) and compute their nearest-neighbor similarity within the same LAION corpus.

The results, summarized in Table 9, show a substantial separation between OOD-Eval prompts and LAION captions. Compared to LAION captions among themselves, OOD-Eval prompts exhibit approximately 47% lower lexical similarity (BM25) and 9% lower semantic similarity (CLIP), indicating that they are meaningfully out-of-distribution with respect to the training data. Notably, the larger gap observed under BM25 aligns with our findings in A.2.1, where we show that semantic embedding models such as CLIP often assign inflated similarity scores to rare or unseen concepts due to coarse semantic grounding. In contrast, BM25 more precisely reflects true distributional distance for rare objects, making it a reliable indicator of OODness in this setting.

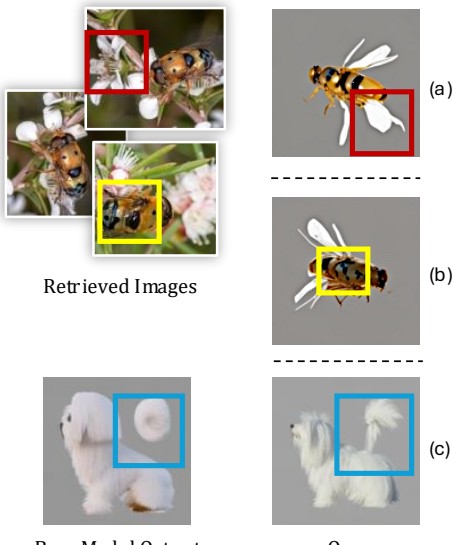

## A.4 User Study Details

We provide additional details about the user study referenced in Tab.3. The study involved 8 different objects, each evaluated using 3 methods: MV-RAG, MVDream, and ImageDream. For each object, participants were first shown a brief text description along with two sample images of the object to establish context. They were then shown sets of four images corresponding to different views-generated by each method. The internal order of the methods was randomized per object to mitigate ordering bias. Participants were asked to rate the following three questions on a scale from 1 to 5: (Q1) "How well do the 4 images match object?", (Q2) "How realistic do the 4 images look overall?", and (Q3) "How well do the 4 images appear to be consistent with each other, as if they show different views of the same 3D object?". The study was conducted using Google Forms, and participants viewed the images on a computer screen. The user population consisted of 30 randomly selected individuals across diverse ages, ethnicities, and genders.

Figure 14: **Limitations.** (a) Visually biased retrieved images (e.g., repetitive white flowers) introduce artifacts in the generated multiviews. (b) The model struggles to reproduce fine-grained textures, such as the hoverfly's dorsal pattern. (c) When the base model (MVDream) is assigned a high attention weight ($\alpha$), 3D structural inaccuracies from the base model (e.g., a floating tail) are inherited.

## A.5 Limitations

While effective, our method has several limitations. It relies heavily on both the quality of the retrieved image corpus and the capabilities of the underlying generative model, MVDream. When the base model lacks prior knowledge of the object and retrieval fails to provide informative or diverse references, the generated multiviews can be inaccurate or implausible.

As shown in Fig. 14, errors may arise when the retrieved images are visually biased-e.g., all showing similar white flowers, leading to reduced diversity and visual artifacts. Furthermore, our training objective promotes texture variation, which can make it difficult to reproduce fine-grained or specific patterns, such as the hoverfly's dorsal markings.

Our model also employs an adaptive mechanism that balances attention between the base model and the retrieval adapters, based on a similarity score between the generated initial views and retrieved images. When the base model demonstrates high similarity to the target object but exhibits 3D structural errors (such as a floating dog tail), these artifacts may be inherited.

Further, while our current implementation generates four views, this is not a fundamental limitation of our retrieval-augmented framework, which could be applied to backbones that produce a larger set of views (e.g., Zero123++ (Shi et al., 2023a)).

Lastly, our method introduces a retrieval phase prior to generation. Although this adds computational cost relative to standard text-to-image pipelines, the overhead is minimal. As shown in Table 8.

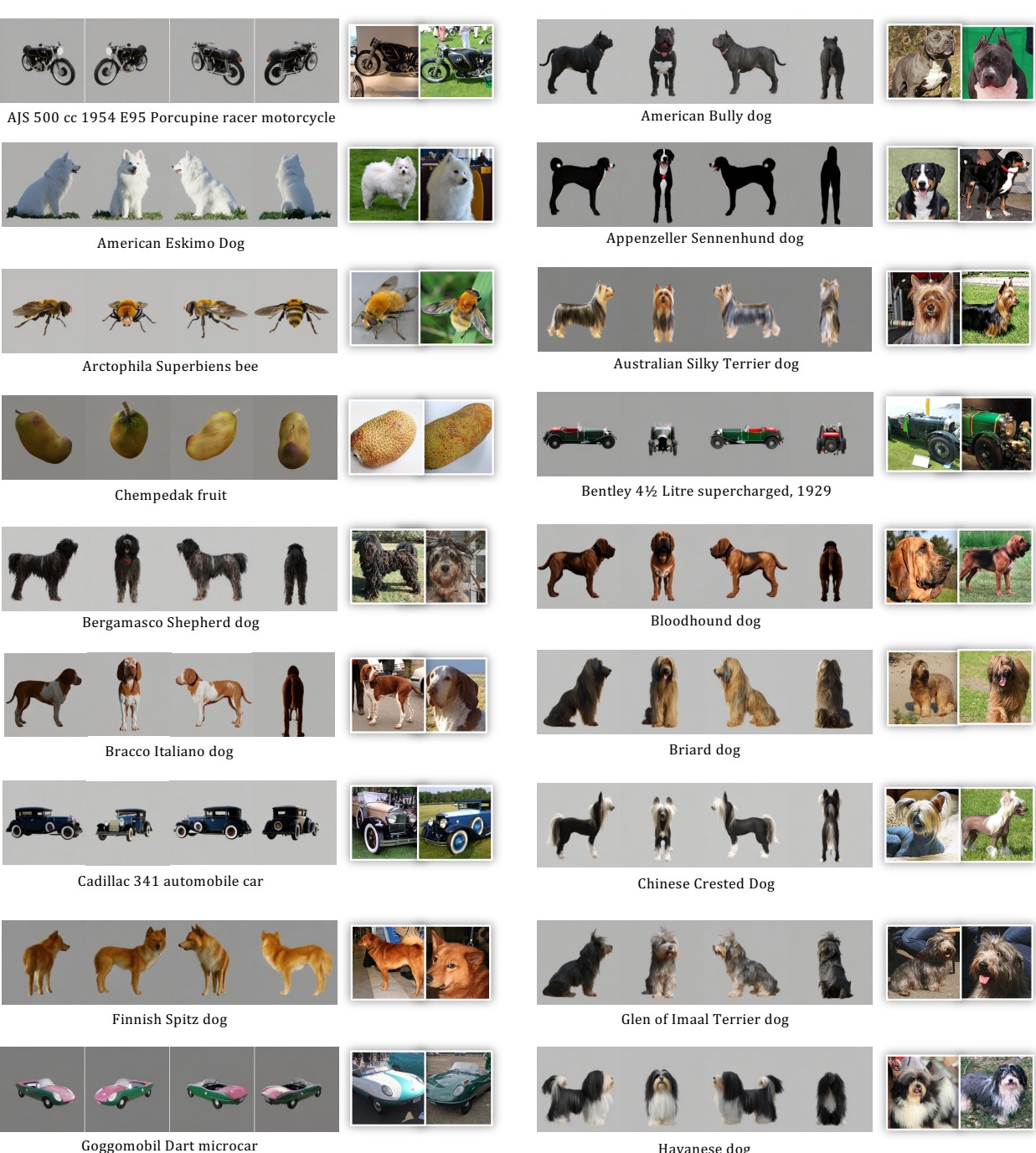

AJS 500 cc 1954 E95 Porcupine racer motorcycle

American Bully dog

American Eskimo Dog

Appenzeller Sennenhund dog

Arctophila Superbiens bee

Australian Silky Terrier dog

Chempedak fruit

Bentley 4½ Litre supercharged, 1929

Bergamasco Shepherd dog

Bloodhound dog

Bracco Italiano dog

Briard dog

Cadillac 341 automobile car

Chinese Crested Dog

Finnish Spitz dog

Glen of Imaal Terrier dog

Goggomobil Dart microcar

Havanese dog

Figure 15: **Additional Results.**

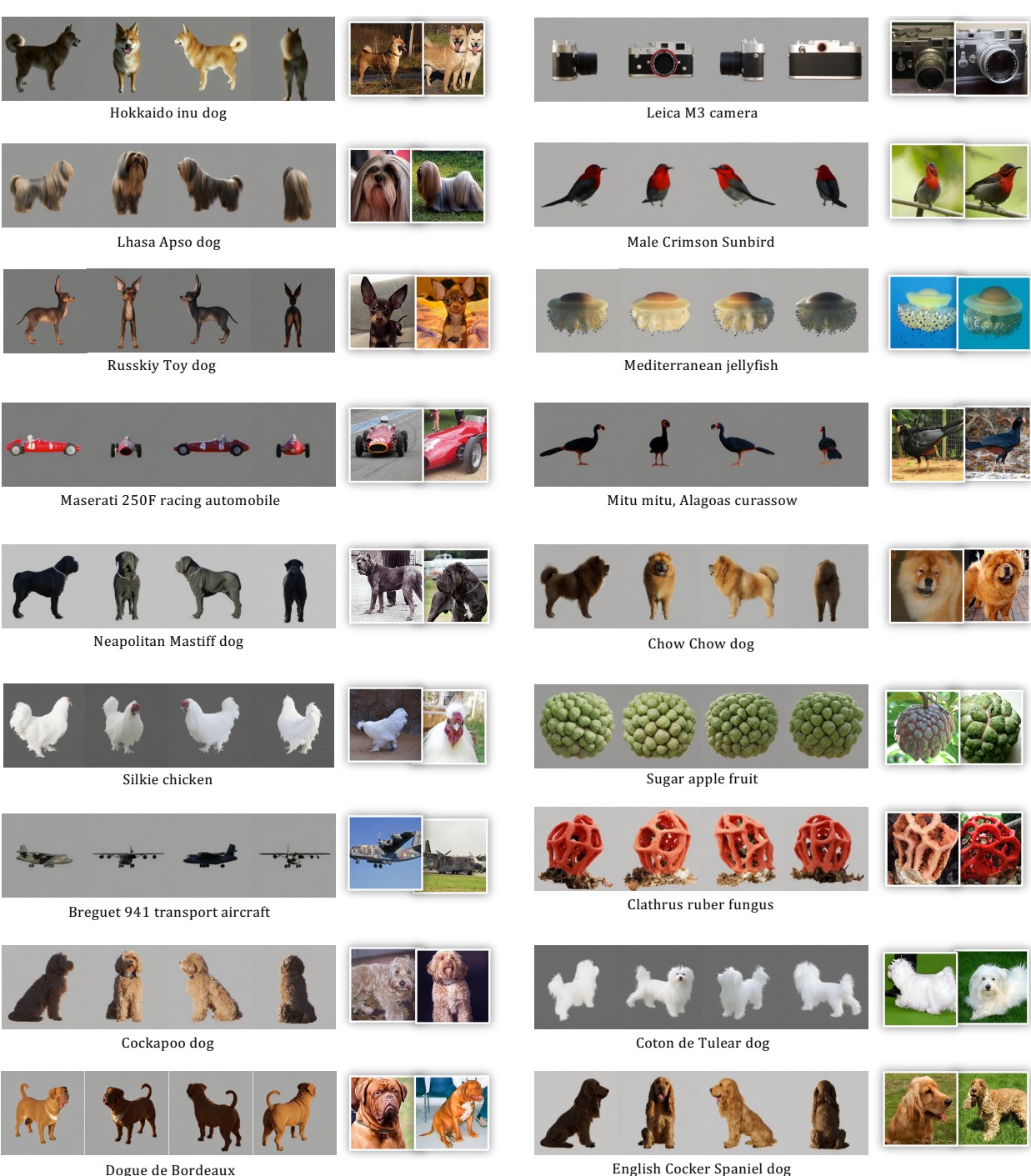

Figure 16: **Additional Results.**

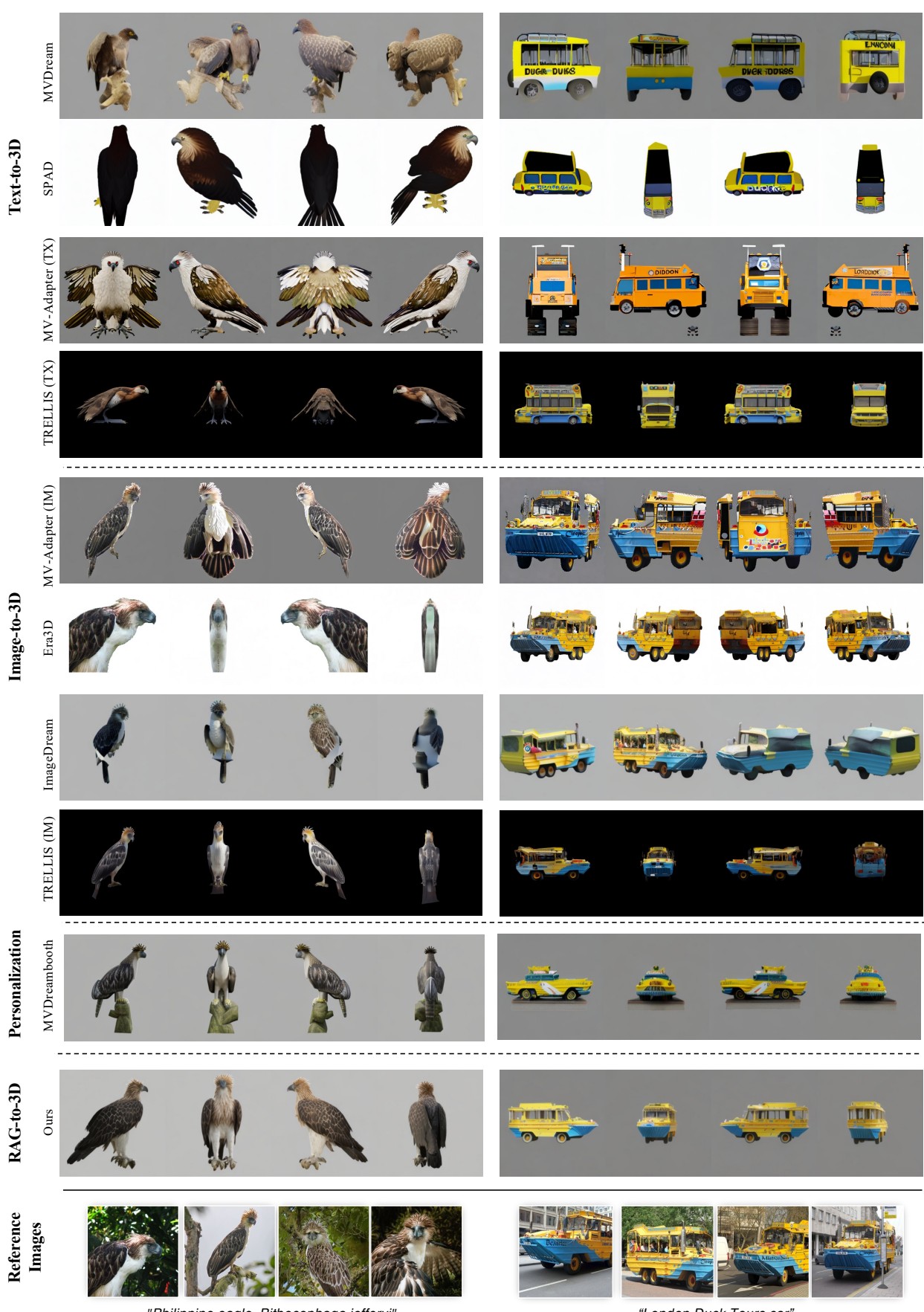

Figure 17: **Additional qualitative evaluation.** Additional examples to those shown in Fig. 5.

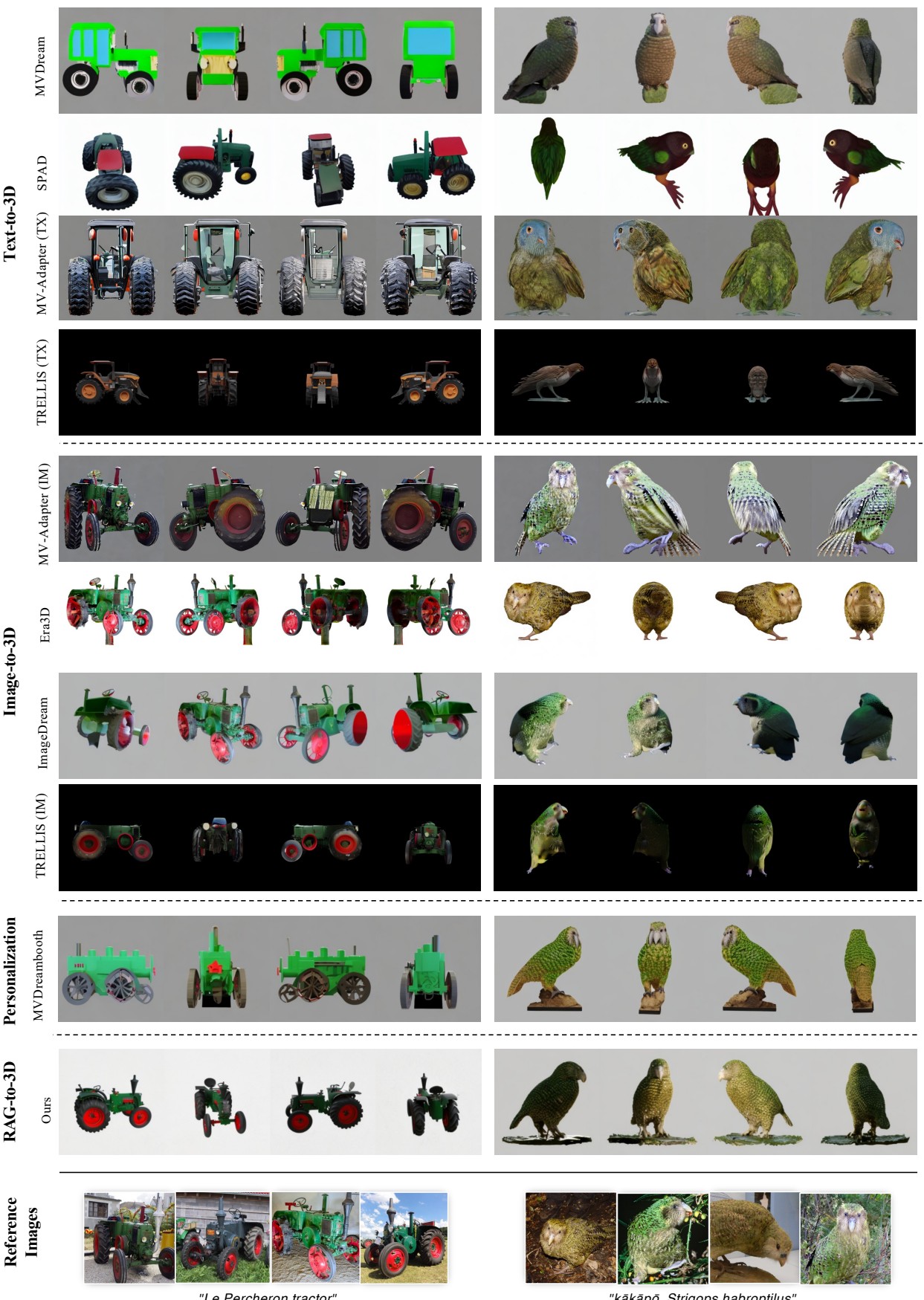

Figure 18: **Additional qualitative evaluation.** Additional examples to those shown in Fig. 5.

