# OpenReview forum: "MV-RAG: Retrieval Augmented Multiview Diffusion"
_TMLR — Under review for TMLR_

### Review · Reviewer_8f81 · 2026-06-25

**Summary Of Contributions:**

The authors study the problem of generating several different 'camera' views of an object described via a text prompt. These views can be used downstream to generate 3D models via Gaussian splatting. They especially evaluate this in an OOD setting, ie for prompts describing images with little training data. Their solution involves doing RAG on the LAION dataset of image - text pairs, finding images that are likely similar to the prompt. By conditioning on those during the generation in the U-Net via a cross-attention mechanism, this allows the model to generate new types of images (ideally). They also develop a training process both with direct multi-view data, and from more general 2D data to ensure diversity.

**Audience:**

Yes

**Audience Explanation:**

As far as I know yes. The RAG method seems novel and quite interesting, and it has (I think) good results. But I'm not from the community so take it with a grain of salt.

**Claims And Evidence:**

Yes

**Claims Explanation:**

I'm very much not an expert in this field. I thought however the paper is quite well written and clear. The model design seems thorough and sensible, and the performance of their model is good (where it matters; namely OOD). There are (from what I can tell) some underspecified details, and I found it a bit annoying that I had to chase the appendix for important details that imo should be in the main text.

**Requested Changes:**

- Please use citep :-) Really hard to read many sentences!
- One thing I was missing for a proper understanding is that 'multiview' generation is then followed by another step to take the multiple views and create a 3D model (If I understood correctly). Imo that can go in the intro for noobs like me :-)
- I had some trouble following these details in the methodology when reading the main text, I think it's better if it's out of the appendix
    - Which parts of the model are pretrained, and what pretrained model is used. Eg I think the MVDream backbone is frozen and existing weights are used from A3.3?
    - What datasets are used for training and RAG. Eg a 'diverse 2D database' is mentioned, but why not just say it's LAION? Quite an important detail imo. And also not so clear what dataset the model is trained on, only the preprocessing is discussed.
- I found the alpha gating thing a bit ad-hoc and not well motivated. First it is mentioned it's used to check whether a small generation is similar enough to the retrieved images. And then regenerate with lower alpha to make it more similar or something. But why not do that to begin with to ensure it's similar anyways? It's also not clear to me how this alpha is computed exactly. I also couldn't find a clear ablation of this mechanism

---

> ### Author Response · Authors · 2026-07-14
>
> We thank the reviewer for their positive assessment, noting that the paper is "well written and clear," the design is "thorough and sensible," and the out-of-distribution (OOD) performance is strong. We appreciate your suggestions to bring key architectural and dataset details forward from the appendix to improve readability. Below, we address your requested changes directly.
>
> 1. **Formatting of Citations:**
>
> We thank the reviewer for pointing this out. We have thoroughly reviewed the manuscript and corrected the citation formats using \citep where appropriate for smoother flow and readability.
>
> 2. **Clarification multiview definition:**
>
> The reviewer is correct: the multiview diffusion model generates a set of 3D-consistent 2D views, which can be subsequently processed by a 3D reconstructor (such as LGM, using Gaussian Splatting or NeRF) to lift them into a continuous 3D representation. We have added this explanation at the beginning of the Introduction to make the notion accessible to all readers.
>
> 3. **Specification of Pre-trained and Frozen Components:**
>
> The reviewer is correct. To specify fully: the base multiview diffusion is MVDream, which remains frozen throughout training to preserve its pre-trained structural and geometric priors. The image encoder CLIP is likewise frozen. The only trainable components are the cross-attention retrieval adapters and the Perceiver Resampler that compresses the retrieved image tokens. We elaborate on the trained and frozen components in Section 3.3 and provide additional specifications such as exact model versions in Appendix A.3.3.
>
> 4. Clarification of Datasets used for Training and Retrieval:
>
> The retrieval database indeed varies by setup, which is why the methodology used general terms, but we agree the specifics are important and now name them explicitly in the main text:
> - **Training:** The 3D training mode uses Objaverse; the 2D training mode uses a subset of ImageNet-21K, chosen because it conveniently groups unposed images by semantic concept (Section A.3.1).
> - **Evaluation:** For OOD evaluation, the retrieval database is our OOD-Eval image set combined with MS-COCO, added to increase retrieval difficulty. For in-domain evaluation, the 3D targets are taken from Objaverse, and retrieval is performed over LAION-400M, which is large enough to contain relevant matches for Objaverse objects (Section A.3.6).
>
> 5. **Motivation and Mechanism of the Adaptive Coefficient ($\alpha$):**
>
> We appreciate the reviewer’s question and would like to clarify the explicit two-step design of this mechanism, why it cannot be done directly from the start, and its core motivation:
> - **The core challenge:** We cannot know a priori how familiar the base model is with a specific prompt. Diffusion models approximate the score function ∇x log p(x|y), so an in-domain prompt naturally guides early denoising toward an accurate concept representation, while an OOD prompt quickly deviates.
> - **Why it is needed:** The base model's large-scale pretraining provides a powerful structural foundation that we want to preserve, but selectively. As shown in Figure 4, for a rare concept where the model recognizes the global structure but not the specific identity (e.g., a specific chicken species), a moderate α retains the robust geometric shape while retrieval injects fine-grained details. Conversely, for completely unfamiliar concepts (Figure 4, top row), a low α prevents the uninformed base prior from corrupting the generation.
> - **How α is computed:** We perform a short 10-step lookahead pass using only the base text attention, then compute the DINOv2 similarity between this initial output and the retrieved images. This similarity serves as a proxy for the base model's confidence: high similarity indicates the concept lies in a high-density region of the prior (favoring high α), low similarity indicates an OOD concept (lowering α toward retrieval). This is precisely why it cannot be set from the start: without the lookahead, there is no empirical signal from which to compute α. Once computed, α is used across all prior-guided attention layers.
> - Ablation: We quantitatively ablated the mechanism by perturbing the estimated α with fixed offsets on OOD-Eval:
>
> | **Metric** | **α−0.5** | **α−0.3** | **α−0.1** | **α (Ours)** | **α+0.1** | **α+0.3** | **α+0.5** |
> |------------|----------:|----------:|----------:|-------------:|----------:|----------:|----------:|
> | **CLIP ↑** | 68.10 | 68.38 | 69.76 | **71.77** | 71.18 | 71.00 | 66.18 |
> | **DINO ↑** | 41.39 | 42.24 | 45.63 | **51.19** | 51.11 | 43.94 | 31.18 |
> | **IR ↑** | 61.44 | 62.12 | 64.64 | **67.41** | 67.21 | 64.11 | 53.21 |
>
> The estimated α consistently outperforms all shifted variants, with large offsets in either direction (over-trusting the prior or over-trusting retrieval) substantially degrading performance. This confirms the per-prompt estimation is both meaningful and well-calibrated around its operating point.

---

### Review · Reviewer_CbEF · 2026-07-03

**Summary Of Contributions:**

The paper proposes MV-RAG, a retrieval-augmented multiview diffusion pipeline for text-to-3D generation, aimed particularly at rare or out-of-domain concepts. Given a text prompt, the method retrieves relevant 2D images from a large in-the-wild corpus and conditions a multiview diffusion model on these images. The key technical components are a retrieval-conditioned cross-attention mechanism, a hybrid training strategy combining 3D multiview data and unposed 2D image collections, a held-out image prediction objective for 2D data, and a prior-guided fusion mechanism that balances the base model prior against retrieved visual evidence.

The main strengths are the clear motivation, the practical use of abundant 2D data rather than scarce 3D assets, strong empirical results on the proposed OOD-Eval benchmark, comparisons against several relevant text-to-3D, image-to-3D, and personalization baselines, and useful ablations. The main weaknesses are that the OOD benchmark is newly curated by the authors and not independently established, the evaluation relies heavily on embedding-based image-image metrics and a re-rendering proxy for 3D consistency, and the user study is relatively small. The method also depends on retrieval quality and the underlying MVDream backbone.

**Audience:**

Yes

**Audience Explanation:**

The paper addresses a timely and important problem at the intersection of retrieval-augmented generation, diffusion models, and 3D content generation. TMLR readers interested in generative modeling, multimodal learning, retrieval-augmented systems, and 3D generation would likely find the approach useful. The idea of using large-scale 2D retrieval to improve rare-concept multiview generation is practically relevant and may inspire follow-up work beyond the specific MVDream backbone.

**Claims And Evidence:**

Yes

**Claims Explanation:**

The main claims are mostly supported. The paper provides quantitative results on OOD-Eval showing consistent gains over strong baselines in CLIP/DINOv2/IR similarity and FID, and the re-rendered evaluation provides additional evidence for improved multiview consistency. The in-domain evaluation suggests that the OOD gains do not substantially harm standard object generation. The ablation studies support the roles of the 2D mode, 3D mode, retrieval count, and retrieval method. Qualitative results are also persuasive.
That said, the evidence has limitations. OOD-Eval is introduced by the authors, so the broader generality of the results is not fully established. The 3D consistency evaluation uses a reconstruction-and-rerendering proxy rather than direct geometric ground truth. The user study covers only a limited subset of objects and methods. I therefore find the claims generally supported, but I would like the authors to make the evaluation limitations more explicit and, if possible, expand the human study or provide additional external validation.

**Requested Changes:**

- Clarify the construction of OOD-Eval and possible sources of bias. The benchmark is central to the paper’s claims, so the authors should more explicitly describe how concepts were selected, how overlap with training data was checked, and whether any manual filtering could favor retrieval-based methods.

- Strengthen or qualify the 3D consistency claim. The current re-rendered evaluation is useful, but it is still a proxy mediated by LGM. Please discuss this limitation clearly and, if feasible, add an additional consistency metric or geometric evaluation.

- Expand the user study or better justify its scope. The current user study uses only 8 objects and compares against a small subset of methods. Since perceptual quality and rare-concept alignment are hard to measure automatically, a broader user study would strengthen the evidence.

- Provide clearer details on retrieval filtering and failure cases. The method depends strongly on BM25 retrieval and thresholding. Please report sensitivity to retrieval quality and give more quantitative evidence for noisy or irrelevant retrievals.

Minor：

- Include more results with modern large-scale text-to-image or text-to-3D systems, even if only through available demos, to better contextualize OOD performance.

- Provide a clearer discussion of whether MV-RAG’s improvements come from retrieved visual identity, retrieved texture, retrieved geometry cues, or simply better object-class grounding.

- Release OOD-Eval and code if possible, or provide enough implementation details to reproduce the retrieval corpus, prompts, and evaluation protocol.

- Improve figure readability in several qualitative comparisons. Some outputs are small and hard to inspect without zooming.

---

> ### Author Response · Authors · 2026-07-14
>
> We thank the reviewer for their thorough evaluation, positive remarks, and constructive suggestions. Below we address all requested changes and minor points directly.
>
> 1. **Clarification on OOD-Eval construction and potential bias**:
>
> - **Concept Curation & Overlap Check:** For rigorous OOD testing, we used GPT-4o to curate long-tail entities absent from standard text-to-3D training sets. To prevent overlap, we filtered out concepts with high semantic/lexical similarity to ImageNet21K and LAION captions. Quantitative distance analysis (Appendix A.3.6) confirms OOD-Eval prompts have ~47% lower lexical similarity (BM25) and ~9% lower semantic similarity (CLIP) than LAION captions, proving they are genuinely OOD.
> - **Category Distribution:** The distribution—Animals (81.1%), Vehicles (12.2%), Food (5.1%), Artifacts (1.0%), and Fungus (0.5%)—reflects real-world long-tail entities. Unique biological species and machinery designations (e.g., "Axolotl", "1998 Fiat Multipla") enable unambiguous text-based RAG querying, whereas furniture is defined by broad attributes. The "Animal" super-category also offers massive geometric diversity (e.g., jellyfish vs. dog). No manual filtering favored retrieval. **OOD-Eval data and code are in the supplementary material.**
>
> 2. **Qualification of the 3D consistency metric (Re-rendering proxy):**
>
> - **Metric protocol:** We lift the 4 generated views to 3D via LGM and render 18 novel views, evaluating them via image similarity against the reference. Inconsistencies among input views degrade the 3D lifting, resulting in blur or distortion in novel views. This follows standard practice (e.g., ImageDream) due to the absence of ground-truth 3D models for arbitrary OOD prompts.
> - **Qualification:** We agree this is a proxy mediated by LGM that measures consistency jointly with reconstruction quality rather than geometry in isolation. Because direct geometric metrics are unfeasible without ground-truth geometry, we complement this proxy with a user study directly evaluating 3D consistency ("How well do the 4 images appear to be consistent with each other..."), where MV-RAG leads significantly.
>
> 3. **Scope of the User Study:**
>
> Our study evaluated 3 axes (Realism, Alignment, Consistency) across 8 objects and 3 models, with 30 participants providing 72 ratings each (>2,000 total judgments). It was restricted to close backbones (MV-RAG, MVDream, ImageDream) to isolate the retrieval mechanism's impact without architectural confounds, capping objects to avoid fatigue. We clarify this in Appendix A.4.
> Beyond this, we compared against CLAY (Rodin), a state-of-the-art commercial system, under the same protocol (Appendix A.2.6, Table 7), where MV-RAG was strongly preferred on all criteria.
>
> 4. **Details on retrieval filtering and failure cases:**
>
> - **Filtering mechanism:** Top $K$ images are retrieved via BM25. A gating mechanism discards images below a threshold (BM25 score = 9.36), yielding $K' \le K$ images. If $K' = 0$, retrieval modules are disabled, safely falling back to the base prior (Appendix A.2.3).
> - **Sensitivity to retrieval quality:** First, performance as a function of $K'$ (Fig. 8) degrades gracefully as fewer images pass, remaining effective even at $K'=1$. Second, Table 5 shows BM25 achieves 0.852 Precision@5 on OOD queries, outperforming CLIP (0.537); Table 6 details why lexical matching excels over semantic embeddings for rare entities.
> - **Failure cases:** Failure occurs when the base model lacks an internal prior and retrieval returns incorrect images that pass the BM25 filter, leaving the model with inaccurate grounding.
>
> **Response to Minor Points**
> - **Contextualization:** Appendix A.2.6 provides comparisons against large-scale systems (TRELLIS and CLAY/Rodin). Using CLAY's public demo, we conducted a user study where MV-RAG was strongly preferred on all criteria (Table 7).
> - **Source of improvements:** Gains stem from visual grounding and fine-grained textures in retrieved images, which compensate for the base model's weak understanding of rare concepts. The base model provides structural layout, while retrieval injects fine features (e.g., Fig. 6(a)). Clarified in Section 4.2.
> - **Code & Benchmark:** The OOD-Eval dataset and code are in the supplementary material and will be fully open-sourced.
> - **Figure Readability:** We increased the resolution of qualitative figures in the revision to ensure legibility.

---

### Review · Reviewer_Cqnc · 2026-07-03

**Summary Of Contributions:**

This paper introduces a novel text-to-3D (Text-to-3D) generation pipeline named MV-RAG, aiming to address the issues of geometric inconsistency and text alignment disparity when existing models generate Out-of-domain (OOD) or rare concepts.
Its main contributions include:
1. It is the first time that the Retrieval-Augmented Generation (RAG) framework has been successfully applied to multiview 3D synthesis, by using relevant 2D in-the-wild images as the conditioning input for the multiview diffusion model.
2. The paper designs a novel 2D-3D hybrid training mechanism. This mechanism combines the reconstruction target on structured multiview data (using augmented conditioning views to simulate retrieval variance) and the "held-out view" prediction target on the 2D image set (inferring 3D consistency without camera poses).
3. A prior-guided attention mechanism is introduced. During the inference stage, the similarity is calculated to dynamically adjust the weight coefficient α of the internal prior in the base model and the external retrieval signal.
4. The paper proposes a new benchmark OOD-Eval with 196 complex prompts, filling the gap in current OOD 3D generation evaluation data.

Strength
1. This paper proposes a complete and well-motivated solution to a key problem in the Text-to-3D field (the generation of OOD concepts).
2. The design of the hybrid training strategy is very ingenious. Especially in the 2D mode, the camera pose is deliberately not provided, forcing the network to directly infer 3D consistency from the unstructured and heterogeneous input, which is conceptually enlightening.
3. Compared with the state-of-the-art baseline models (such as MVDream, ImageDream, TRELLIS, etc.), there is a significant improvement in 3D consistency, photorealism, and text adherence on the OOD-Eval benchmark, without sacrificing the generation performance in the standard domain (in-domain).

Weaknesses
1. The generation quality of this method is highly dependent on the quality and diversity of the 2D retrieval database. When the retrieved images are visually biased (such as highly similar backgrounds and colors), it may lead to artifacts in the output or a decrease in diversity.
2. In the prior-guided attention mechanism, if the similarity score of the base model (such as MVDream) for a certain concept is overestimated, but its 3D structure itself has defects (such as a floating tail), MV-RAG may inherit these 3D structural errors.
3. The proposed benchmark contains only 196 prompts. Given the vast long-tail distribution of rare concepts, this small scale may limit the statistical significance of the quantitative results. Expanding the dataset would yield a more robust evaluation.

**Audience:**

Yes

**Audience Explanation:**

Researchers in the fields of computer vision, graphics, and generative AI will find this paper interesting.
1.The paper discusses how to utilize large-scale 2D data to enhance the generalization ability of 3D models, which is the core challenge in current 3D content creation.
2.This paper innovatively adapts the popular RAG concept in NLP and 2D image generation to multi-view/3D scenes, providing a potential method for addressing cross-modal conditional injection.

**Broader Impact Concerns:**

The model is highly dependent on in-the-wild 2D image libraries, which makes it prone to injecting the inherent social or cultural biases of the network into the generated 3D models. At the same time, if its powerful ability to generate rare concepts is abused, there is a risk of infringing copyright or generating dangerous/deceptive 3D assets. It is recommended that the authors supplement or refine the Broader Impact and ethical considerations statement.

**Claims And Evidence:**

Yes

**Claims Explanation:**

1. On the newly constructed OOD-Eval and standard test sets, through multiple metrics such as CLIP, DINOv2, etc., as well as the rigorous Re-rendered test, it objectively proves its significant advantages in 3D consistency and photorealism.
2. A large number of visual comparisons provided demonstrate the generation advantage for rare concepts, and in the user study, its realism, text alignment, and 3D consistency scores are significantly ahead of existing baseline models.
3. Through comprehensive ablation experiments, it rigorously proves the necessity and effectiveness of core designs such as the 2D/3D hybrid training strategy, the data augmentation mechanism, and the specific number of retrieved images.

**Requested Changes:**

To further enhance the quality of this paper, the author is advised to consider the following optional adjustments:

1. It is suggested to add a sensitivity analysis regarding the fusion weight λ of the text features and the retrieval features during the hybrid training stage, as well as the hyperparameter λ' during the inference stage.
2. In Appendix A.5, some limitations are discussed. It is recommended to further demonstrate the smoothness of the decline of the adaptive fusion coefficient α when falling back to the base model.

---

> ### Author Response · Authors · 2026-07-14
>
> We thank the reviewer for their constructive feedback, encouraging summary, and insightful suggestions. We are glad they found our hybrid training strategy "ingenious" and "conceptually enlightening," noting our significant improvements across OOD benchmarks. Below, we address specific concerns and requested changes.
>
>
> 1. **Generation dependency on the quality and diversity of the 2D retrieval database (Weakness 1):**
>
> We agree, and this failure mode is explicitly analyzed in the paper. **Section A.2.2** presents a controlled experiment on retrieval variance comparing three regimes: low variance (highly similar images) yields accurate but less diverse generations and can propagate instance-specific bias; our default moderate variance (top-k) offers the best balance of quality and diversity; and high variance can challenge 3D structure recovery, though results remain superior to baselines. The artifact case the reviewer describes, visually biased retrievals with similar backgrounds and colors, is shown in **Figure 14(a)** (repetitive white flowers) and discussed in **Limitations (A.5). Figure 17** further illustrates both regimes qualitatively (low variance: "London Duck Tours car"; high variance: "Philippine eagle, Pithecophaga jefferyi"). We view this dependency as inherent to retrieval-augmented generation, and our gating mechanism (**A.2.3**) mitigates extreme cases by falling back to the base prior when retrievals are unusable.
>
>
> 2. **Inheriting 3D structural errors from the base model when $\alpha$ is overestimated (Weakness 2):**
>
> The reviewer is correct: if the adaptive fusion coefficient $\alpha$ overestimates the base model's proficiency, intrinsic structural defects (e.g., a floating tail) can be inherited. We highlight this failure mode in the Limitations section (A.5, Figure 14c). Importantly, the error originates in the base model's prior rather than in our architecture. A key advantage of MV-RAG is its modularity: as stronger multiview generative models emerge (e.g., Zero123++, or SOTA systems such as TRELLIS), they can serve as backbones in our pipeline, directly mitigating these inherited base-level artifacts.
>
>
> 3. **Evaluation scale of the OOD benchmark (Weakness 3):**
>
> We appreciate the suggestion to expand the dataset. We curated OOD-Eval to target rare, long-tail concepts strictly absent from standard datasets, making prompt collection highly rigorous. Notably, OOD-Eval’s size (196 prompts) aligns with or exceeds established text-to-3D benchmarks like **GPTEval3D (110), MATE-3D (160), and T3Bench (300)**. Moreover, our margins over the strongest baseline are large and consistent across all metrics (e.g., +11.6 DINOv2), providing robust evidence of improvement at this benchmark size.
>
>
> 4. **Sensitivity analysis of the fusion weights $\lambda$ (training) and $\lambda'$ (inference):**
>
> **Training ($\lambda$):** During training, conditioning is $f = \lambda \cdot f_{\text{txt}} + f_{\text{ret}}$. Since the base model's text projections are frozen, down-weighting them is standard in adapter tuning (e.g., IP-Adapter, Ye et al., 2023): suppressing the dominant text signal allows gradients to flow into the new retrieval adapters. Consistent with Sec. 3.3, small $\lambda$ values ease adaptation of the retrieval branch.
>
> **Inference ($\lambda'$):** During inference, formulation shifts to $f = \alpha \cdot f_{\text{txt}} + (\lambda' - \alpha) \cdot f_{\text{ret}}$, where $\lambda'$ regulates the total conditioning energy. While performance remains stable within a reasonable window, excessively increasing $\lambda'$ introduces high-frequency visual artifacts and over-saturation due to excessive signal magnitude, whereas decreasing it leads to overly smoothed textures lacking fine-grained detail.
>
>
> 5. **Smoothness of the decline of $\alpha$ when falling back to the base model:**
>
> We appreciate this opportunity to clarify the transition dynamics of $\alpha$:
> - **Continuous, smooth transition**: $\alpha$ is driven by a continuous DINOv2 similarity score between initial lookahead views and retrieved images. It varies smoothly as a concept shifts from out-of-domain to in-domain, yielding a stable blend of features rather than an abrupt switch. **Figure 4** demonstrates this empirically: outputs vary gradually across the $\alpha$ spectrum, and the effect saturates smoothly at high $\alpha$, where results are nearly indistinguishable from the base model. Appendix A.2.4 discusses failure modes of misestimated $\alpha$.
> - **Discrete lower bound**: If zero retrieved images pass our filtering threshold, we default to the base model, preventing conditioning on noise and securing a performance floor.
>
>
> 6. **Broader Impact Concerns:** We added a Broader Impact Statement in Section 6 to explicitly address social/cultural data bias, copyright concerns, and mitigations like input content filtering.